# TYK2 mediates neuroinflammation in Alzheimer's disease brains with TDP-43 pathology

Laura E. König [1,2,12], Steve Rodriguez[1,2,12], Clemens Hug [1], Shayda Daneshvari[1,2], Alexander Chung [1,2], Mark Appleman [1,2], Max Tsai[1,2], Gary A. Bradshaw[1], Asli Sahin[2], Yuyu Song [1,2], George Zhou [2], Robyn J. Eisert [1,8], Federica Piccioni [3,9], Christine Marques[2,4], Sharon Powley [2], James Yarmolinsky[5,6], Brian J. Wainger [2], Sudeshna Das [2,4], Marian Kalocsay [1,10], Abbas Dehghan [5,6], Ioanna Tzoulaki [5,6,7], Artem Sokolov [1,11], Peter Sorger [1], David E. Root [3] & Mark W. Albers [1,2] ✉

Neuroinflammation is a pathological feature of neurodegenerative diseases like Alzheimer's disease and ALS. Cytoplasmic dsRNA (cdsRNA) triggers a type-I interferon response in human neural cells, leading to their death, and is found in neurons of *C9ORF72*-ALS patients. Here, we report the spatial coincidence of cdsRNA and pTDP-43 inclusions in human postmortem tissue with Alzheimer's disease pathology, and upregulated interferon response genes in affected regions. CdsRNA also accumulates in a human TDP-43 G298S iPSC cortical neuronal model. We use cryptic exon detection as a proxy for TDP-43 mis-localization and demonstrate that FDA-approved JAK inhibitors baricitinib and ruxolitinib, which block interferon signaling, show protective effects only in brains with elevated cryptic exon expression. A CRISPR screen reveals *TYK2* as a top hit, and *TYK2* knockdown and the selective TYK2 inhibitor deucravaci-tinib rescue cdsRNA-induced toxicity. We find parallel neuroinflammatory mechanisms, dependent on TYK2 - a potential disease-modifying target - for TDP-43-associated Alzheimer's disease and *C9ORF72*-ALS.

Efficacious therapies based on principles of precision medicine have impacted millions of patients suffering from heterogeneous diseases such as cancer[1–3]. In recent years, mounting evidence indicates that clinical dementia, like cancer, is also a collection of multifaceted neurodegenerative diseases[4–6]. With the number of dementia patients increasing in our aging society[7], it is likely that personalized treatments for specific subsets of dementia will be important to address the huge unmet need for effective therapy[8].

[1]Laboratory of Systems Pharmacology, Harvard Program in Therapeutic Science, Harvard Medical School, Boston 02115 MA, USA. [2]Department of Neurology, Sean M. Healey & AMG Center for ALS, Massachusetts General Hospital, Charlestown, MA, USA. [3]Broad Institute of MIT and Harvard, Cambridge, MA, USA. [4]Harvard Medical School, Boston, MA, USA. [5]Dementia Research Institute, Imperial College London, London, UK. [6]Department of Epidemiology and Bios-tatistics, School of Public Health, Imperial College London, London, UK. [7]Biomedical Research Foundation, Academy of Athens, Athens, Greece. [8]Present address: The Department of Biological Chemistry and Molecular Pharmacology, Harvard Medical School, Boston, MA, USA. [9]Present address: Merck Research Laboratories, Cambridge, MA, USA. [10]Present address: Department of Experimental Radiation Oncology, The University of Texas MD Anderson Cancer Center, Houston, TX, USA. [11]Present address: Etiome, Cambridge, MA, USA. [12]These authors contributed equally: Laura E. König, Steve Rodriguez. ✉e-mail: albers.mark@mgh.harvard.edu

Alzheimer's disease (AD), a disease defined by amyloid beta (Aβ) plaques and neurofibrillary tangles (NFTs) of hyperphosphorylated tau protein, can be associated with other proteinopathies, including cytoplasmic inclusions of phosphorylated TAR DNA-binding protein 43 (pTDP-43)[9]. TDP-43 pathology is observed in up to 57% of AD patients[10] and is one of the most prominent proteinopathies after Aβ and tau pathology[11]. TDP-43 is a nuclear protein involved in RNA processing[12]. Ectopic cytoplasmic inclusions of this protein are thought to deprive the nucleus of TDP-43 function and dysregulate RNA processing and expression[13–18]. For instance, loss of function of TDP-43 triggers derepression of immunostimulatory genomic double-stranded RNA (dsRNA) that can activate pattern recognition receptors (PRRs)[19,20], which results in increased cell autonomous and non-cell autonomous inflammatory processes[21]. Previously, we demonstrated that cytoplasmic pTDP-43 inclusions are spatially coincident with cytoplasmic dsRNA (cdsRNA) in amyotrophic lateral sclerosis (ALS) and frontotemporal dementia (FTD) patients with an intronic expansion of hexanucleotide repeats in the *C9ORF72* gene[22]. Independent from TDP-43 pathology, cdsRNA triggers an interferon-mediated toxic innate immune response in a sequence-independent fashion and leads to neuronal cell death in in-vitro and mouse models[22]. Elucidating the role of cdsRNA in other neurodegenerative diseases causing dementia, such as AD, could offer a paradigm to target neuroinflammatory mechanisms using principles of precision medicine.

Here, we show that immunogenic cdsRNA and cytoplasmic pTDP-43 inclusions are spatially coincident in brains with coexisting AD pathology as well as in a cell-based model of TDP-43 pathology, and demonstrate robust type-I interferon (IFN-I)-signaling in AD patients' brains. We updated our machine learning pipeline for drug repurposing in AD (DRIAD-SP)[23] to include cryptic exon (CE) expression, which is a proxy for TDP-43 pathology[24–29]. Here, we demonstrate a protective signal for baricitinib and ruxolitinib, both FDA-approved Janus kinase (JAK) inhibitors that block the interferon response, in only a subset of AD patients with elevated CE expression. To evaluate the mechanism of action of neuroprotective JAK inhibitors, we conducted a genome-wide CRISPR screen that reveals that knockout of *TYK2*, a JAK kinase family member, robustly rescues cdsRNA-induced toxicity. In addition to baricitinib and ruxolitinib, we also found that deucravacitinib – a selective TYK2 inhibitor recently approved for moderate-to-severe plaque psoriasis[30] – exhibits a neuroprotective effect in three different neural cell models with more potency compared to the other JAK inhibitors. Finally, we analyze potential inflammatory biomarkers for dsRNA-mediated neuropathology by using cell-based assays as well as conducting a human observational study of plasma levels in individuals with a TYK2-activity-reducing single-nucleotide polymorphism (SNP), and find that CXCL10 is a candidate biomarker for validation in clinical trials.

## Results

### Presence and immunogenicity of cdsRNA in AD
To assess the presence of cdsRNA in neurons of patients with AD pathology, we stained formalin-fixed paraffin-embedded (FFPE) amygdala sections of human postmortem brains of healthy individuals ($n = 9$) and of patients who met the pathologic diagnosis of AD ($n = 10$) (Fig. 1a; Supplementary Table 1). We stratified the patients and controls based on the severity of pTDP-43 pathology and detected cdsRNA in all individuals with severe pTDP-43 pathology by immunostaining. In contrast, whenever pTDP-43 inclusions were absent, our immunostainings were negative for cdsRNA as well (Supplementary Fig. 1a,b). In individuals with mild pTDP-43 pathology, we detected cdsRNA in 33% of cases. Next, we evaluated the spatial coincidence of cdsRNA and pTDP-43 by performing cyclic immunofluorescence (CyCIF)[31], which enabled multiplexed staining of pTDP-43 inclusions, cdsRNA, NFTs, Aβ plaques, and nuclei on the same tissue section (Fig. 1b, lower panel of healthy control and AD, respectively). We compared the acquired images to Luxol fast blue-hematoxylin and eosin (LHE) stainings as well

as immunohistochemical stainings of cdsRNA, pTDP-43, tau, and Aβ (Fig. 1b, upper panel of healthy control and AD, respectively). While NFTs and Aβ plaques were detected in the tissue, neither pathology overlapped spatially with cytoplasmic pTDP-43 inclusions nor cdsRNA (Fig. 1b). However, pTDP-43 and cdsRNA were spatially coincident (enriched two-fold in AD cases; Supplementary Fig. 1c), paralleling our previous observations in ALS brains[22]. If this cdsRNA acts as a damage-associated molecular pattern (DAMP) in neurons, then we anticipate it would trigger an IFN-I response through the activation of the PRRs RIG-I and MDA5 (Supplementary Fig. 1d). To test whether cdsRNA is immunogenic in brains with AD pathology, we immunostained amygdala sections for phosphorylated protein kinase R (PKR), another PRR that binds to cdsRNA and activates autophosphorylation[32] (Supplementary Fig. 2a). We observed phosphorylated PKR (pPKR) only in cdsRNA-positive AD brains, close to 66% of it being coincident with cdsRNA (Supplementary Fig. 2b), indicating a potential direct link between cdsRNA and the activation of PRRs.

Furthermore, we wanted to see if we could replicate the observation of increased cdsRNA levels in a more controlled setting of TDP-43 pathology. For this purpose, we used a recently established human induced pluripotent stem cell (iPSC) line that harbors a pathogenic TDP-43 mutation (TDP-43^{+/G298S})[33] and also shows increased interferon signaling[34]. As a positive control for cdsRNA detection, we lipofected control iPSC-derived NGN2 cortical-like neurons with poly(I:C), a dsRNA mimetic consisting of polyinosinic-polycytidylic acid, and were able to detect an increase in cdsRNA signal (Fig. 1c). We then differentiated an isogenic pair of the mutant iPSCs into NGN2 cortical-like neurons[34] and observed a significant increase in cdsRNA in cells with the familial TDP-43 mutation compared to control cells (Fig. 1d). These observations identify shared characteristics of human AD-brain cells and a smaller, better controlled model systems for TDP-43 pathology. Together, these findings show a link between immunogenic cdsRNA and TDP-43 mislocalization in our model system and in AD patients with TDP-43 pathology.

### Increased interferon signaling in AD
Next, we conducted a differential gene expression analysis of RNA-sequencing data from five relevant brain regions available from the Religious Order Study/Memory and Aging Project (ROSMAP)[32] and Mount Sinai Brain Bank (MSBB)[35] AMP-AD databases to test our hypothesis of elevated interferon signaling. We first identified differentially expressed genes between AD patients and healthy controls, then assessed enrichment of a preselected interferon-stimulated gene (ISG) signature relative to gene sets from the KEGG Medicus database[36], which contains pathway gene sets for normal and diseased states. The ISG gene set ranked among the top 3 most enriched gene sets across all brain regions examined (Fig. 2a). Among the other top 15 pathways, many were related to ISG signaling, including antigen processing and presentation by MHC Class II and TLR-NFκB signaling (Fig. 2b). Within the ISG set, we found 139 unique ISGs that were significantly upregulated and 51 ISGs that were significantly downregulated (adjusted $P$-value < 0.05) in AD patients compared to healthy controls. The parahippocampal gyrus (PHG), a brain region that shows atrophy early on in disease progression[37,38], had the greatest number of upregulated ISGs, followed by the dorsolateral prefrontal cortex (DPFC) and the inferior frontal gyrus (IFG). In addition, we observed upregulation of ISGs, to a lesser extent, in the superior temporal gyrus (STG) and frontal pole (FP; Fig. 2c; Supplementary Data 1).

### DRIAD-SP predicts efficacy of blocking interferon signaling
To examine whether interferon signaling in brains with AD pathology was associated with pTDP-43 pathology, we extended our DRIAD-SP framework along two axes. We previously published a machine learning framework for Drug Repurposing In Alzheimer's Disease with Systems Pharmacology (DRIAD-SP)[23] that links the prediction of

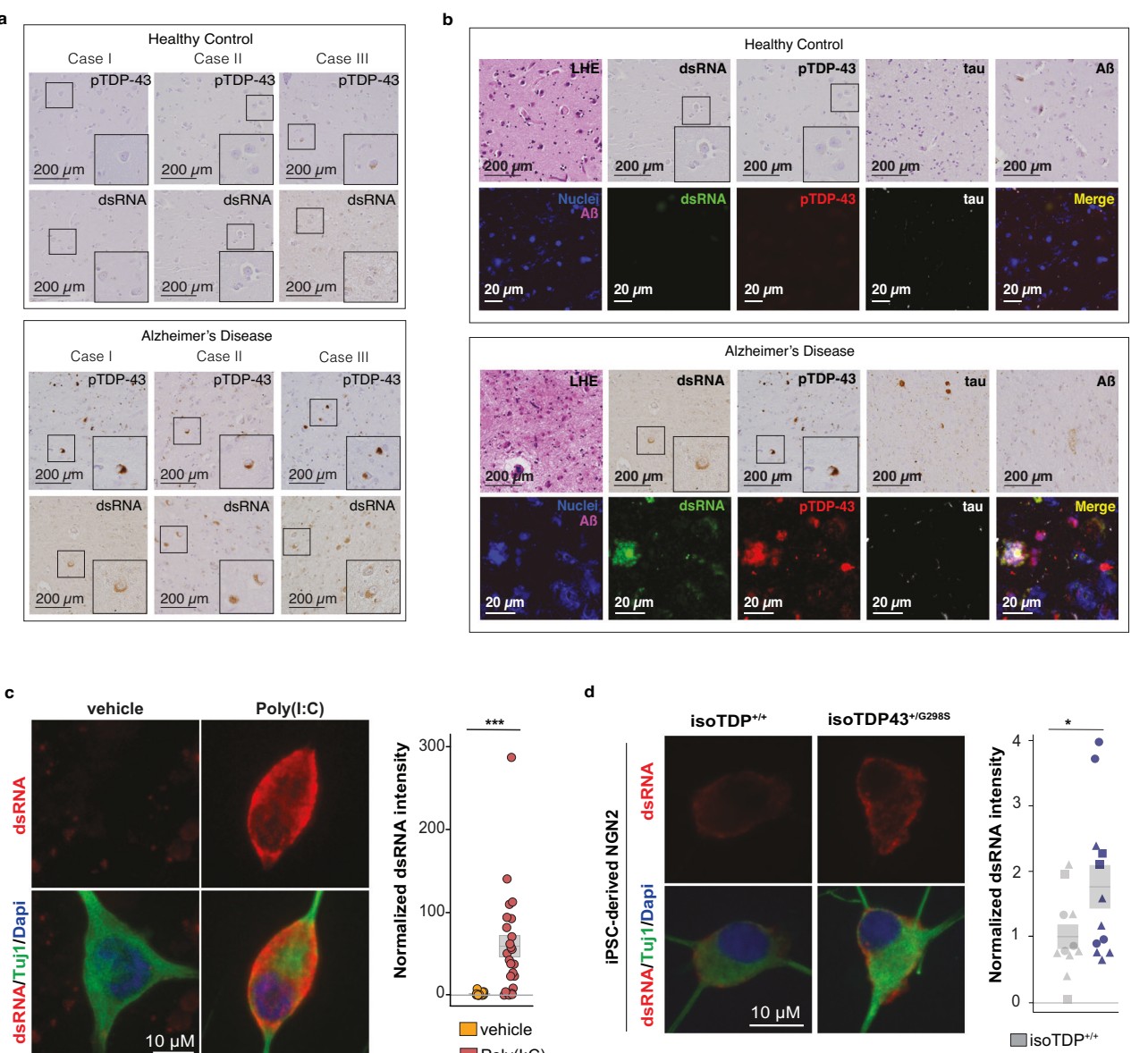

**Fig. 1 | CdsRNA is spatially coincident with pTDP-43 inclusions in AD.**
**a** Immunohistochemistry (IHC) for cdsRNA and pTDP-43 inclusions in human postmortem brain sections of the amygdala. Top panel: healthy control cases, bottom panel: AD patient cases. Zoom-ins with 2x magnification. This experiment included nine controls brains and ten AD patients (Supplementary Fig. 1a). **b** Luxol fast blue-hematoxylin and eosin (LHE) staining, IHC, and cyclic immuno-fluorescence (CyCIF) of human postmortem brain sections of the amygdala. IHC for cdsRNA (green), pTDP-43 (red), tau (white), and Aß (purple) on brain sections of the same healthy control (top panel, healthy control case II)/AD patient (bottom panel, AD case I) as Fig. 1a. CyCIF panels show nuclei (Sytox Blue, blue), Aß, dsRNA, pTDP-43, and tau as well as a merge of all channels. CyCIF was done on a single brain section of the same healthy control as the IHC images, but of a different AD patient (also see Supplementary Table 1). This experiment included nine controls brains (one used for multiplexed staining) and ten AD patients (two used for multiplexed staining; Supplementary Fig. 1a). **c** Immunofluorescent staining of wildtype

(isoTDP-43[+/+]) differentiated iPSC-derived NGN2 cortical-like neurons transfected with poly(I:C) as a positive control showing dsRNA (K1 antibody, red), Tuj1 (neuronal marker, green), and nuclei (DAPI, blue) (left). Quantification of the normalized dsRNA/K1 intensity (right), $P = 0.0001695$. Each dot is a field of view (40x magnification, one well analyzed per condition). **d** Immunofluorescent staining of wild-type (isoTDP-43[+/+]) and mutant (isoTDP-43[+/G298S]) differentiated iPSC-derived NGN2 cortical-like neurons showing dsRNA (K1 antibody, red), Tuj1 (neuronal marker, green), and nuclei (DAPI, blue) (left). DsRNA signal intensity normalized to dsRNA signal intensity in the isoTDP-43[+/+] control. Quantification of the normalized dsRNA/K1 intensity (right) with $P = 0.04319$. Each dot is a well (40x magnification, 2/3 well analyzed per condition). Each shape is a different differentiation batch (3 differentiation batches analyzed). Center line, median; box limits, upper and lower quartiles. Statistical tests: unpaired two-tailed Student's $t$-test. No adjustments for multiple comparisons. Patient information of depicted brains is available in Supplementary Table 1. Source Data is provided as a Source Data file.

disease stage (early- vs. late-stage AD based on the Braak AD staging system[39]) to drug-induced molecular signatures in human neural cells. Importantly, it does not predict the direction of a drug's effect – that is, whether the drug rescues or exacerbates disease phenotypes – but rather nominates promising candidates for further in-vitro and in-vivo validation[23]. Through this pipeline, we previously identified baricitinib,

ruxolitinib, and tofacitinib, which are FDA-approved JAK inhibitors that inhibit interferon signaling, as drug candidates for repurposing in AD[23] (Supplementary Fig. 3a).

First, we improved DRIAD-SP's predictive power by training and evaluating predictors of Braak disease stage in a linear manner using ordinal ridge regression (Fig. 3a), instead of the

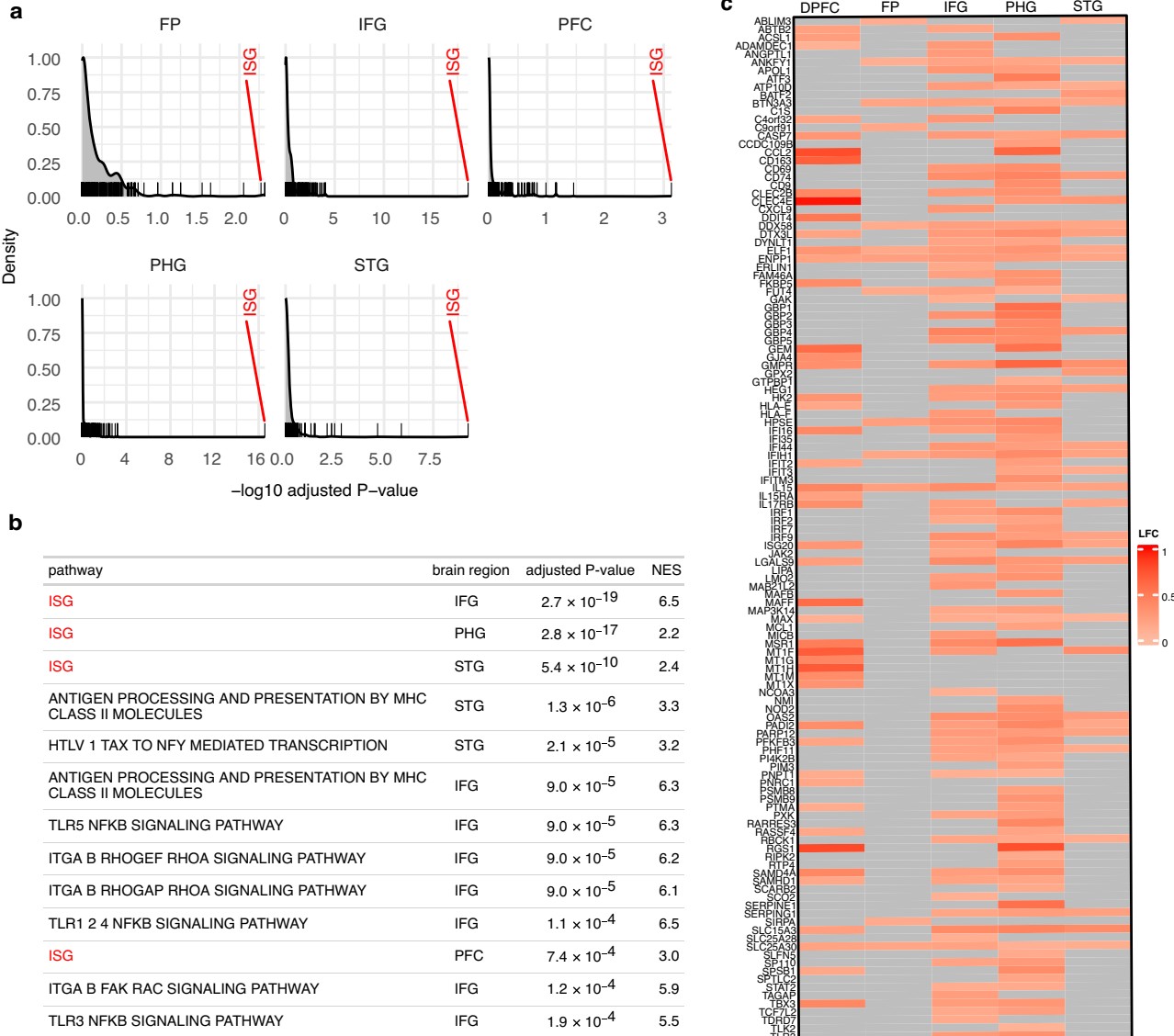

**Fig. 2 | Upregulation of interferon-stimulated genes (ISGs) within brain regions relevant in AD. a** Gene set enrichment analysis results of genes differentially expressed in Alzheimer's Disease (AD) patients vs. healthy controls using RNA-seq data derived from the ROSMAP[32] and MSBB[35] databases. The distribution in gray and tick marks is enrichment of gene sets from the KEGG Medicus database[36]. Marked in red is the enrichment of a custom ISG list. Enrichments were computed using the fgsea R package[115]. **b** Table showing the top 15 most enriched gene sets across brain regions. Enrichments were computed using a one-sided GSEA test implemented in the fgsea package[115]. *P* values were corrected for multiple testing using the Benjamini-Hochberg procedure. **c** Heat map of RNA-sequencing data derived from the ROSMAP[32] and MSBB[35] databases comparing AD patients and healthy controls. Differential expression shown in log2 fold change (LFC) for pre-selected ISGs. Full data set in Supplementary Data 1. NES Normalized Enrichment Score, DPFC dorsolateral prefrontal cortex, FP frontal pole, IFG inferior frontal gyrus, PHG parahippocampal gyrus, STG superior temporal gyrus.

binary classification, as was done previously. This approach allows for training on the full scale of stage I to stage VI along the Braak spectrum[39], without the need to stratify patients into "early" and "late" categories. Second, we used the expression of CEs as a proxy for the presence of pTDP-43 inclusions (Fig. 3b). Previous work demonstrated that the pre-mRNAs of stathmin-2 (STMN2)[16–18] and UNC13A[14,15] have binding sites for TDP-43 and that TDP-43 represses CE inclusion in the canonical mRNA of both STMN2 and UNC13A during splicing. pTDP-43 pathology is associated with nuclear hypofunction of TDP-43, which leads to CEs being included in the processed mRNA, resulting in a short variant of STMN2 mRNA[16–18] (STMN2 short) due to premature polyadenylation, and two versions of CE-inclusive UNC13A mRNA, which we refer to as UNC13A-CE1 and UNC13A-CE2[14,15]. Numerous studies demonstrated the expression of CE-containing mRNA and de novo proteins also in AD[24–29]. Based on these findings, we posited the presence of these CEs in AD brains as a proxy for TDP-43 pathology. As a proof of concept, we tested our computational

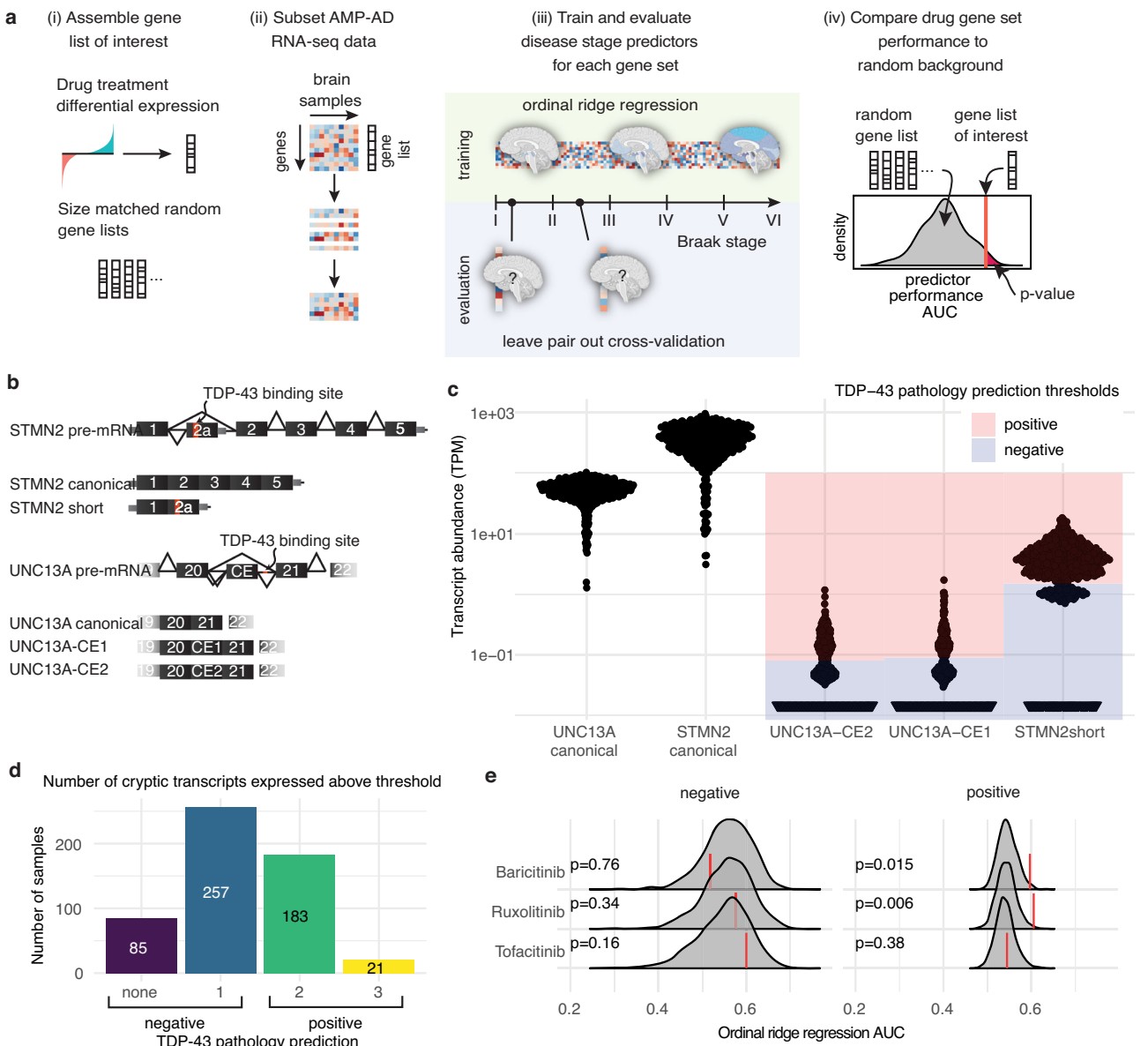

**Fig. 3 | Prediction of drug efficacy in patients stratified by TDP-43 pathology.**
**a** Schematic representation of the DRIAD-SP drug efficacy prediction pipeline. From left to right (i) gene sets are assembled from differential gene expression analysis of cell lines treated with drugs of interest against DMSO controls. Random gene sets of matching sizes are generated for comparison. (ii) RNA-sequencing data from AD brain samples is subset to only contain genes that are present in the gene set that is being tested. (iii) The RNA-sequencing subset is used to fit ordinal ridge regression models predicting Braak disease stage of the patient cohorts. Leave pair out cross-validation[123] is used to evaluate model performance (areas under the curve; AUC). (iv) Drug efficacy is predicted by comparing the performance of the model using the drug gene sets to the size-matched random gene sets. **b** Schematic representation of TDP-43 pathology-associated splice variants that include cryptic exons (CEs) in STMN2 and UNC13A. **c**, Expression of TDP-43 associated splice variants in the posterior cingulate cortex (PCC) brain tissues of the ROSMAP[32] patient cohort. Blue shaded regions are below, and red regions are above the chosen threshold for expression to be considered positive. **d** Per-patient-quantification of the number of CE transcripts that are expressed above the chosen thresholds. Patients are predicted to be TDP-43 pathology-negative if they express one or fewer of the CE transcripts, and positive if they express two or more. **e** Performance of DRIAD-SP models for three selected drugs. Drug efficacy was assessed separately in patient populations according to their predicted TDP-43 pathology. Red lines indicate the DRIAD-SP model performance trained on the drug gene sets, whereas the gray shaded regions correspond to the distribution of model performances based on random gene sets. Empirical one-sided *P* values were computed as the fraction of random background gene sets with higher AUC values than the drug-derived gene set. *P* values were corrected for multiple testing using the Benjamini-Hochberg procedure.

approach by checking for the presence of CEs in RNA-sequencing data[40] from FACS-sorted single neuronal nuclei from the neocortex of deceased ALS patients, as about 97% of ALS patients exhibit pTDP-43 inclusions[41]. We confirmed that neuronal nuclei with loss of nuclear TDP-43 into cytoplasmic aggregates have higher reads of the CEs STMN2 short (log2 fold change (LFC) = 3.55; *P* < 0.0001), UNC13A-CE1 (LFC = 4.20; *P* = 0.0002) and UNC13A-CE2 (LFC = 4.05; *P* = 0.0006) relative to control neuronal

nuclei that contain TDP-43 (Supplementary Fig. 3b). By contrast, the number of reads of canonical STMN2 were reduced to a smaller degree (LFC = −1.76; *P* = 0.0017) and not reduced for UNC13A (LFC = 0.278; *P* = 0.11) mRNAs in TDP-43-deficient neuronal nuclei.

With this validation, we stratified patients' RNA-sequencing data derived from ROSMAP and MSBB AMP-AD databases based on the presence of CEs. First, we set the threshold for the presence or absence

of each individual CE above the lowest non-zero peak (essentially > 1 CE transcript detected) in the abundance histograms (transcripts per million TPM > 1.3 STMN2 short; TPM > 0.09 for UNC13A-CE1; TPM > 0.08 UNC13-CE2; Fig. 3c). We classified patients with two or more CE transcripts as TDP-43 pathology-positive (Fig. 3d). In the MSBB data, the counts for UNC13A-CE1 and UNC13A-CE2 were much lower relative to the ROSMAP data. We reasoned that the single-end sequencing protocol to generate the MSBB RNA-sequencing data was less sensitive for CE detection, relative to the paired-end reads in the ROSMAP data (Supplementary Fig. 4a). Consequently, we restricted our analysis to ROSMAP sequencing data from the posterior cingulate cortex (PCC), a brain region impaired early on in disease progression and therefore highly relevant for our study[42]. Of 546 total patients, no CEs were detectable in 85 patients, and one type of CE was detected in 257 patients; 183 patients had two types of CEs, and 21 patients had all three CEs present (Fig. 3d). Hence, 37% of cases were classified "TDP-43-positive"-AD patients, which is a conservative prediction considering that the clinical incidence of TDP-43-pathology in AD patients can reach 57%[10]. Still, we wanted to ensure that our stratification is not compromised by false positives in case of a natural predisposition of these CEs – especially UNC13A-CE1 and UNC13A-CE2 – to coincide. For this purpose, we mapped out CE expression in all 546 analyzed cases and could not identify any form of link in their occurrence (Supplementary Fig. 4b). To determine whether predicted TDP-43 positivity was associated with later Braak stages, we used an ordinal regression model and found no significant difference in Braak staging ($P = 0.27$) between "positive" and "negative" cases (Supplementary Fig. 5a), indicating that CE production was orthogonal to the stage of tau progression.

We assessed the predicted efficacy of baricitinib, ruxolitinib, and tofacitinib in AD patients with two or more CEs (as a proxy of TDP-43 dysfunction) relative to AD patients with one or no CEs (Fig. 3e; further replicates in Supplementary Fig. 5b). Both baricitinib and ruxolitinib, but not tofacitinib, were significant hits in the predictor trained on RNA-sequencing data from AD patients with two or more CEs. On the other hand, none of the three drugs (baricitinib $P = 0.76$, ruxolitinib $P = 0.34$, tofacitinib $P = 0.16$) were significant hits in the predictor trained on no / low CE-expressing AD patients. These findings and the observation that the expression of ISGs is significantly increased in cases that we deemed TDP-43-positive based on cryptic exon expression (Supplementary Fig. 5c) support our hypothesis that cdsRNA associated with TDP-43 dysfunction contributes to IFN-I-mediated neuroinflammatory processes occurring in AD patients' brains.

## CRISPR screen and validation of TYK2

To replicate cdsRNA-mediated inflammation and further determine that cdsRNA is not only immunogenic but indeed triggers neural cell death, we transfected differentiated ReN VM cells with poly(I:C). These differentiated human neural cells contain markers for neurons, astrocytes, and oligodendrocytes, but not microglia[43] and are not sensitive to other common stressors such as DNA-damaging agents and reactive oxygen species (Supplementary Fig. 6), perhaps because this cell line does not express STING[43]. In contrast to our previous work ("One Pot" Differentiation; Supplementary Fig. 7a)[22], we found that differentiating neural cells in a separate dish and then transferring differentiated cells into the assay plate reduced variability between technical replicates ("Separate Pot" Differentiation; Supplementary Fig. 7b). Both baricitinib and ruxolitinib rescued human neural cell death in this refined workflow with greater reproducibility and lower variance with nearly complete rescue at 1 μM and >100% at 10 μM – presumably due to preventing toxicity from the transfection reagent lipofectamine 2000 present in the control. Both drugs significantly inhibited the phosphorylation of STAT1 in a dose-dependent manner (Supplementary Fig. 8a-f) as well as the downstream transcription of ISGs[23].

Since the $EC_{50}$ values of baricitinib ($EC_{50} = 135.9$ nM) and ruxolitinib ($EC_{50} = 152$ nM) are much greater than their affinities to their

target proteins (baricitinib: $IC_{50}(JAK1) = 5.9$ nM, $IC_{50}(JAK2) = 5.7$ nM[44]; ruxolitinib: $IC_{50}(JAK1) = 3.3$ nM, $IC_{50}(JAK2) = 2.8$ nM[45]), we hypothesized that the inhibition of other kinases may be responsible for their neuroprotective effect (Supplementary Fig. 8g). To test this hypothesis, we conducted a genome-wide CRISPR screen to rescue cdsRNA-induced toxicity in differentiated human neural cells[22,46]. We used the Brunello library, which consists of four single guide RNAs (sgRNAs) per human gene and 1000 control sgRNAs[47] (Fig. 4a) expressed in barcoded lentiviral particles to transduce ReN VM cells in their neural progenitor state. After a week-long differentiation, one replicate was harvested to identify genes that were essential for ReN cell differentiation and were not present at the initiation of either poly(I:C) or lipofectamine. Cells carrying sgRNA targeting eight different genes were significantly depleted post-differentiation (DGAT2, SIGLEC11, GSTA2, ABI1, FBXL8, AHCY, C7orf55-LUC7L2, ARHGEF12; Supplementary Data 2), indicating that these genes may be essential for neural cell survival and differentiation. These post-differentiation cells were treated for 48 hours with poly(I:C) or mock lipofectamine transfection, were harvested, genomic DNA was extracted, and the sgRNA sequences within the gDNA were PCR-amplified and sequenced to identify which sgRNA barcodes are enriched. Enriched sgRNAs in the post-poly(I:C) treatment arm versus the mock transfection arm correspond to candidate target genes whose loss-of-function enhances survival. Target genes were considered a hit if the LFC was equal to or greater than 2.0 after rounding to the first decimal place.

We found interferon receptor alpha and beta subunit 2 (IFNAR2; average LFC = 2.23825, $P = 0.0001$), interferon regulatory factor 9 (IRF9; average LFC = 2.0625, $P = 0.0002$), and TYK2 (average LFC = 1.95, $P = 0.0004$) as the best candidates for neural cell rescue upon knockout, whereas knockout of phospholipase C gamma 1 (PLCG1; average LFC = −1.9875, $P = 0.0002$), hepatocyte growth factor-regulated tyrosine kinase substrate (HGS; average LFC = −1.775, $P = 0.0004$), and SQSTM1 (p62, average LFC = −1.7475, $P = 0.0005$) markedly increased the toxicity induced by poly(I:C) (Fig. 4b). A gene enrichment analysis of the top 100 hits using Enrichr[48-50] and the Reactome Pathway database[51] shows that these hits are strongly linked to IFN-I signaling which underlines yet again that cdsRNA toxicity is mediated by an interferon response (Fig. 4c).

Next, we independently validated IFNAR2 and TYK2 as suitable targets to achieve neuroprotection. To evaluate IFNAR2, we treated ReN VM-derived neural cells with an anti-IFNAR2 antibody before introducing poly(I:C) into the cells (Fig. 4d). The blocking antibody significantly rescued dsRNA-mediated neuronal death, but the response was not dose-dependent and plateaued at about 50% rescue (no antibody $P < 0.0001$; antibody 1:50 $P = 0.0001$; 1:25 $P = 0.0002$; 1:10 $P = 0.0011$). This is consistent with our observation that activating IFNAR2 with interferon-α is not sufficient to trigger human neural cell death. We treated cells with interferon-α at a range of doses. After 48 hours, there was no decrease in cell viability at any dose tested (Fig. 4e).

Subsequently, we assessed TYK2 as a potential key player in neuroprotection by generating a knockdown (KD) model in ReN VM cells. Both undifferentiated ($P = 0.0003$) and differentiated ReN VM-derived neural cells show an effective KD of TYK2 (Fig. 4f,g). Toxicity elicited by poly(I:C) is rescued up to 86% in TYK2 KD cells compared to wildtype cells ($P < 0.0001$), presumably short of 100% rescue because of some residual TYK2 expression (Fig. 4h).

## Validation with deucravacitinib, a specific TYK2 inhibitor

To further evaluate TYK2, we used deucravacitinib, a potent small molecule inhibitor[30] that is highly specific for TYK2 over other JAK kinase family members ($IC_{50} = 0.2$ nM compared to $IC_{50}(JAK1-3) > 10$ μM)[52-56] (Supplementary Fig. 8g). We observed that exposure to deucravacitinib rescues dsRNA toxicity fully at all previously tested concentrations from 0.01 μM to 10 μM and that dose-dependent rescue

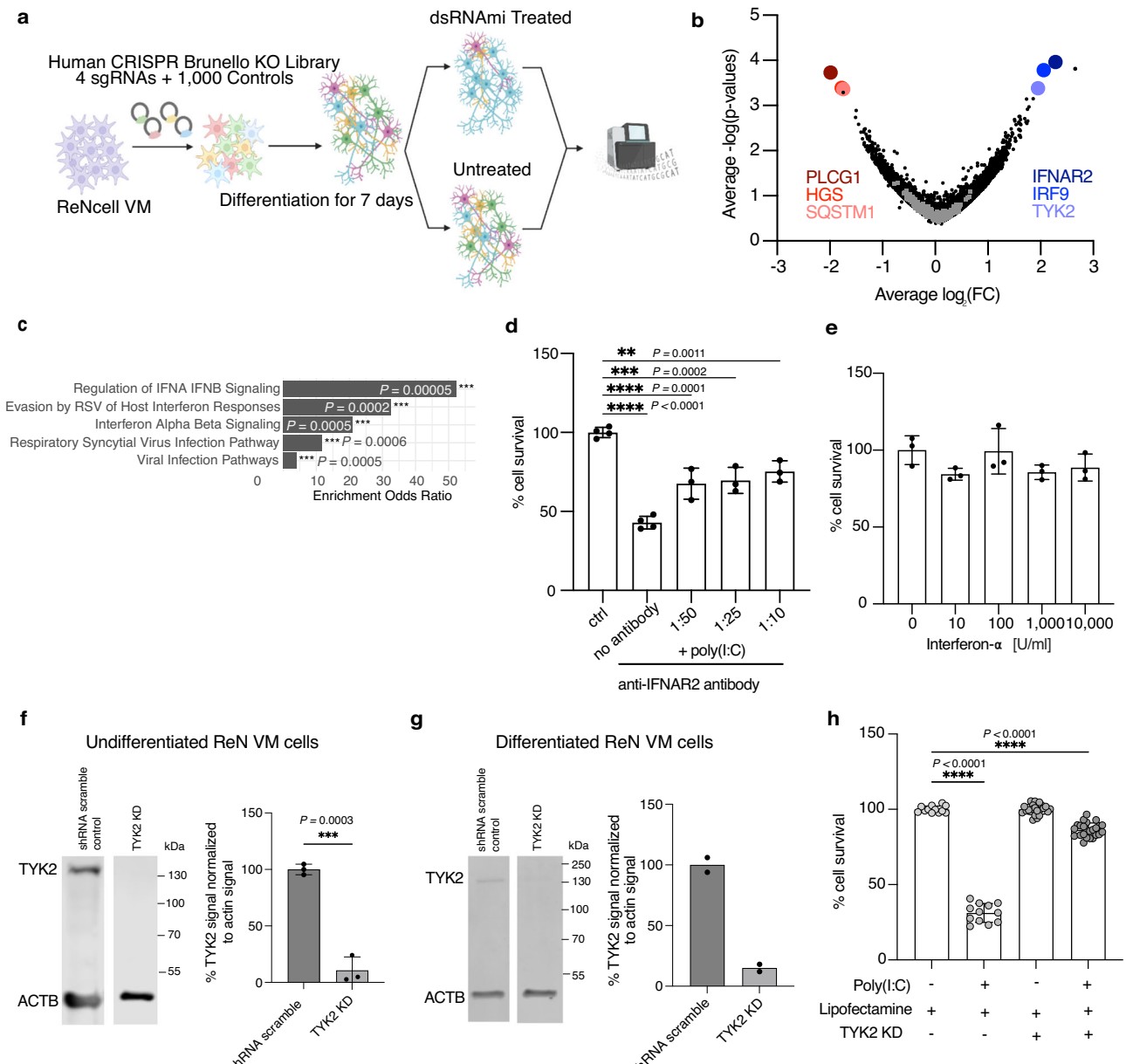

**Fig. 4 | CRISPR screen and subsequent validation identify TYK2 as a therapeutic target. a** Schematic of the CRISPR screen. Created in BioRender. Daneshvari, S. (2026) {https://BioRender.com/a1gztyu}. **b** Volcano plot with log2 fold change (LFC) of genes targeted in the CRISPR screen against their -log(*P*-value). Most abundant knockouts (blue): *IFNAR2* (LFC = 2.23825, *P* = 0.0001), *IRF9* (LFC = 2.0625, *P* = 0.0002), *TYK2* (LFC = 1.95, *P* = 0.0004). Least abundant knockouts (red): *PLCG1* (LFC = −1.9875, *P* = 0.0002), *HGS* (LFC = −1.775, *P* = 0.0004), *SQSTM1* (LFC = −1.7475, *P* = 0.0005). Gray dots represent negative control sgRNAs. P values were calculated using either a negative binomial model with a rank inclusion threshold and permutation-based null distribution, or a hypergeometric model. Full data set in Supplementary Data 2. **c** Reactome Pathway enrichment analysis[48–51] of top 100 hits in CRISPR screen using a one-sided Fisher's exact test. *The five most enriched pathways shown*. **d** Cell survival of ReN VM-derived neural cells treated

with anti-IFNAR2 antibody (no antibody *P* < 0.0001, *n* = 4; 1:50 *P* < 0.0001, *n* = 3; 1:25 *P* = 0.0002, *n* = 3; 1:10 *P* = 0.0011, *n* = 3) after transfection with poly(I:C) or lipofectamine (ctrl, *n* = 4). **e** Cell survival of ReN VM-derived neural cells treated with interferon-a (*n* = 3 each). Image and quantification of Western blot of TYK2 in undifferentiated (**f**; *n* = 3, *P* = 0.0003) and differentiated (**g**; *n* = 2) control and *TYK2* knockdown (KD) ReN VM-derived neural cells normalized to beta actin (ACTB). Samples derive from the same experiment and were processed in parallel. **h** Cell survival of control (*n* = 12 each) and *TYK2* KD (*n* = 24 each) ReN VM-derived neural cells transfected with poly(I:C) or lipofectamine (all *P* < 0.0001). Control cells express scramble shRNA. All replicates are biological replicates. All graph bars represent mean values, while the error bars indicate the standard deviation. Statistical tests (**d**–**h**): ordinary one-way ANOVA with correction for multiple comparisons by Dunnett's testing. Source Data is provided as a Source Data file.

is achieved at significantly lower concentrations (10pM − 10 nM) than baricitinib and ruxolitinib (Fig. 5a; all *P* < 0.0001) with an EC$_{50}$ = 1.483 nM. Deucravacitinib was also found to inhibit STAT1 phosphorylation (downstream protein of TYK2) in a dose-dependent manner (Fig. 5b; Supplementary Fig. 8e, f; all *P* < 0.0001). We validated these results in differentiated ReN CX cells (Fig. 5c, d), a cell line that

shows the same characteristics as ReN VM cells but is derived from the cortex instead of the ventral mesencephalon and generates glutamatergic neurons rather than dopaminergic neurons, and in SH-SY5Y cells, a neuroblastoma cell line that unlike ReN cells expresses STING (Fig. 5e, f; Supplementary Fig. 8h). In both cell types, full rescue of cdsRNA-mediated toxicity was achieved with a 1 μM dose of

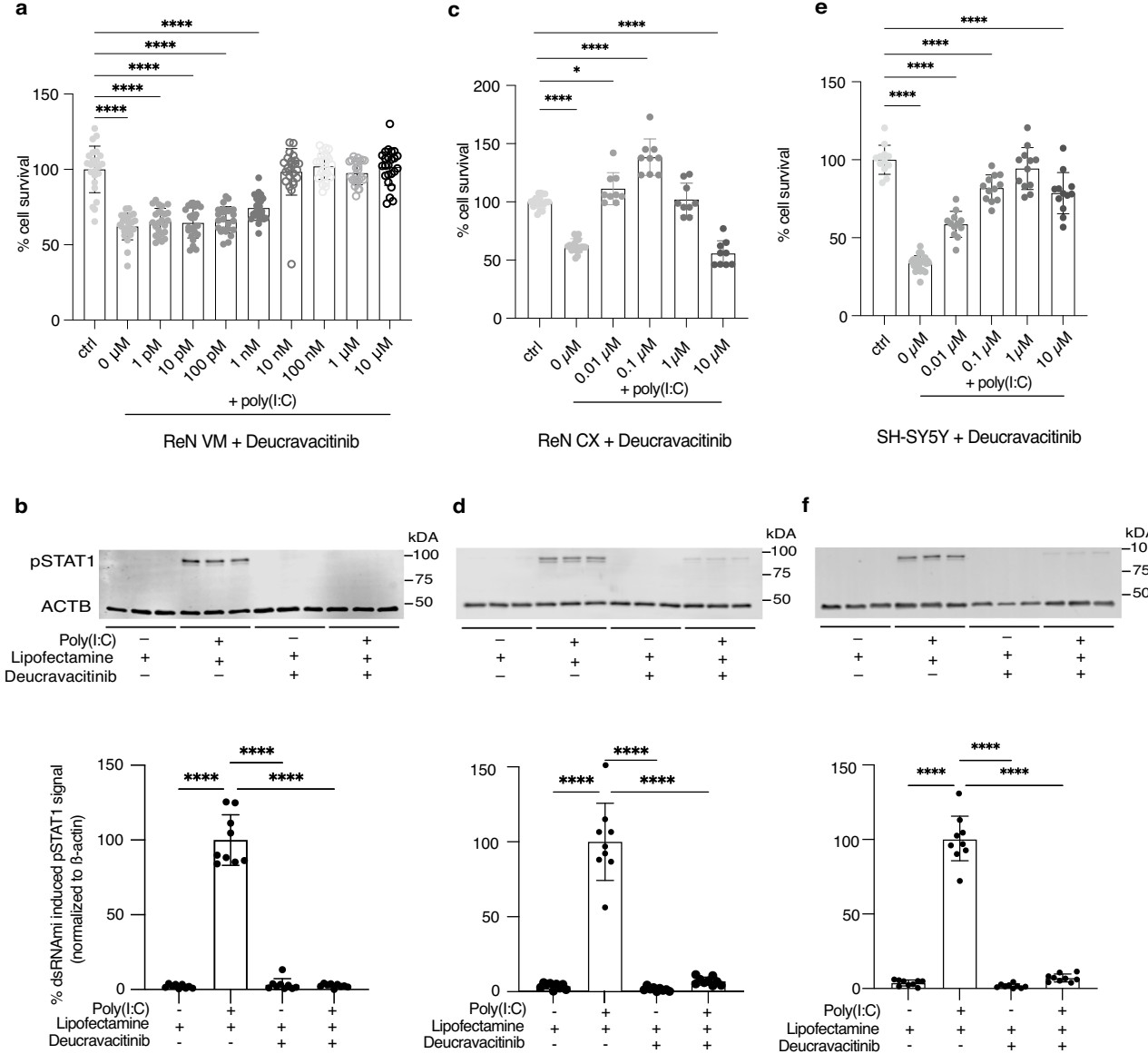

**Fig. 5 | TYK2-selective inhibitor deucravacitinib as a tool compound for TYK2 validation. a** Cell survival of ReN VM-derived neural cells pre-treated with deucravacitinib (0 μM–1 nM $P < 0.0001$; 0.01 μM $P = 0.9987$; 0.1 μM $P = 0.9929$; 1 μM $P = 0.9753$; 10 μM $P = 0.9983$) and afterwards transfected with poly(I:C) or lipofectamine (ctrl). All $n = 24$. **b** Image (top) and quantification (bottom) of Western blot of pSTAT1$^{Y701}$ in ReN VM-derived neural cells 24 hours after treatment with 10 μM deucravacitinib and transfection with poly(I:C) normalized to the housekeeping protein beta actin (ACTB). Top: $n = 3$. Bottom: $n = 9$ (all $P < 0.0001$). Cell survival of ReN CX cell-derived neurons (**c**) and SH-SY5Y cells (**e**) pre-treated with deucravacitinib (CX cells: 0 μM $P < 0.0001$, $n = 18$; 0.01 μM $P = 0.0408$, $n = 9$; 0.1 μM $P < 0.0001$, $n = 9$; 1 μM $P = 0.9868$, $n = 9$; 10 μM $P < 0.0001$, $n = 9$; SH-SY5Y: 0 μM $P < 0.0001$, $n = 24$; 0.01 μM $P < 0.0001$, n = 12; 0.1 μM $P < 0.0001$, $n = 12$; 1 μM $P = 0.4446$, $n = 12$; 10 μM $P < 0.0001$, $n = 12$) and afterwards transfected with poly(I:C) or lipofectamine (ctrl, CX cells $n = 18$, SH-SY5Y $n = 12$). CX cells: EC$_{50}$(deucravacitinib) = 9.424 nM; SH-SY5Y: EC$_{50}$(deucravacitinib) = 16.21 nM. Image (**d** and **f**, top) and quantification (bottom) of Western blot of pSTAT1$^{Y701}$ in ReN CX cell-derived neurons (**d**) and SH-SY5Y cells (**f**) 24 hours after treatment with 1 μM deucravacitinib and transfection with poly(I:C) normalized to the housekeeping protein beta actin (ACTB). Top: $n = 3$. Bottom: $n = 9$ (all $P < 0.0001$). All replicates are biological replicates. All graph bars represent mean values while the error bars indicate the standard deviation. Statistical tests: ordinary one-way ANOVA with correction for multiple comparison by Dunnett's testing. For western blots, samples derive from the same experiment and gels/blots were processed in parallel. Source Data are provided as a Source Data file.

deucravacitinib. Higher concentrations of deucravacitinib seem to be toxic to these cells.

We then applied deucravacitinib to the previously established ReN VM *TYK2* KD cells and observed that adding as little as 0.01 μM deucravacitinib rescued the remaining cell viability deficit caused by residual interferon signaling through the few TYK2 protein that escaped the KD (Fig. 6a). Assessing the phosphorylation and

therefore activation of downstream STAT1 mirrors these results (Fig. 6b).

To further evaluate TYK2 as a key signaling kinase for IFN-I signaling, we performed a deep proteomics analysis from ReN VM-derived neural cells that were treated with baricitinib and deucravacitinib at high doses (10 μM) by quantitative multiplex tandem mass tag (TMT) mass spectrometry (Fig. 6c). Here, we saw

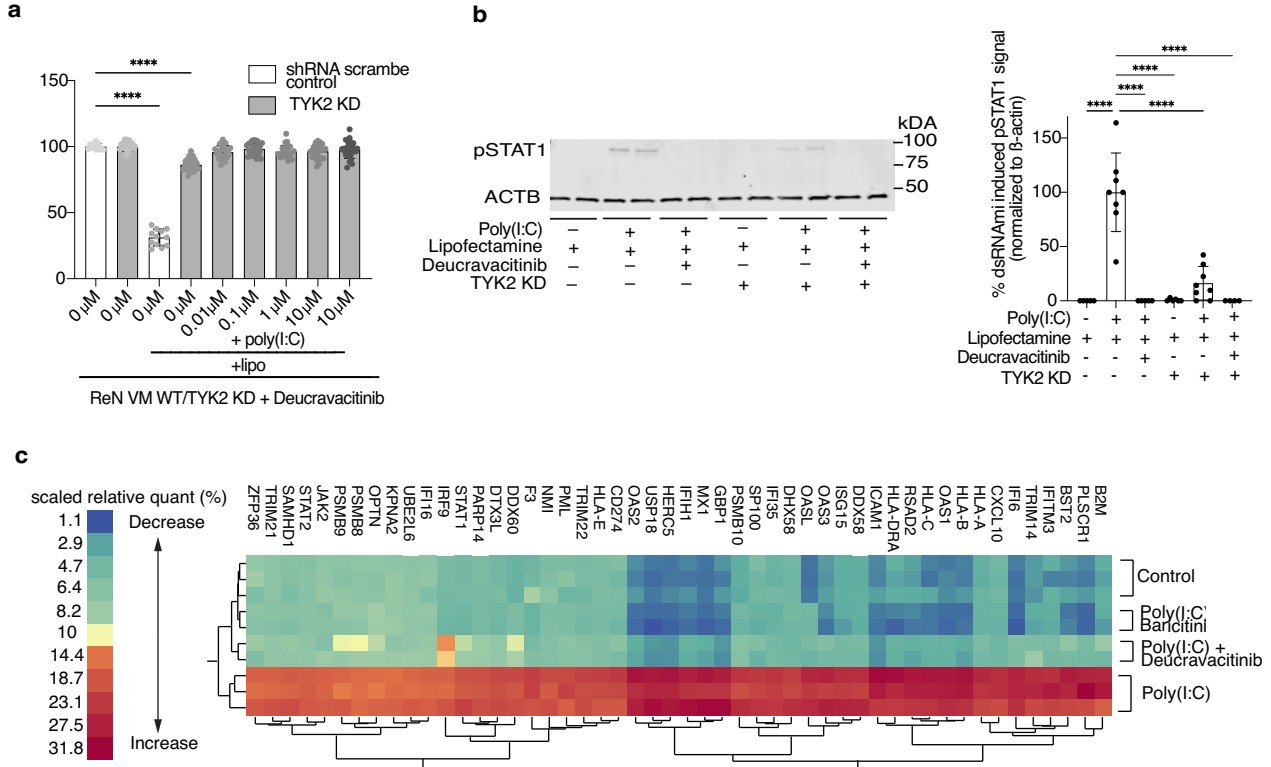

**Fig. 6 | Characterizing TYK2 inhibition with systems pharmacology. a** Cell survival of control (n = 12) and *TYK2* knockdown (KD) (n = 24 each) ReN VM-derived neural cells pre-treated with deucravacitinib (P < 0.0001 where significant) and afterwards transfected with poly(I:C) or lipofectamine as a vehicle control. Control cells express scramble shRNA. **b** Image (top) and quantification (bottom) of Western blot of pSTAT1$^{Y701}$ in control and *TYK2* KD ReN VM-derived neuronal cells 24 hour after treatment with 10 μM deucravacitinib and transfection with poly(I:C) normalized to the housekeeping protein beta actin (ACTB). ROUT (Q = 1%) outlier test was performed, and outliers were excluded. Control cells express scramble shRNA. Top: n = 2. Bottom: n = 8 before outlier test (all P < 0.0001). Samples derive from the same experiment, and gels/blots were processed in parallel. **c** Two-way hierarchical clustering of relative protein abundance of IFN-I-related proteins acquired through TMT multiplex mass spectrometry. Control = lipofectamine (vehicle control). Conditions with drug: n = 2, conditions without drug: n = 3. All replicates are biological replicates. All graph bars represent mean values, while the error bars indicate the standard deviation. Statistical tests: ordinary one-way ANOVA with correction for multiple comparisons by Dunnett's test. Source Data is provided as a Source Data file.

that all quantified ISGs were upregulated by poly(I:C), and this ISG induction was reversed by both baricitinib and deucravacitinib. Taken together, our results indicate that the toxic innate immune reaction to cdsRNA is dependent on TYK2. Furthermore, the action of TYK2 appears to be more complex than just mediating the IFN-I pathway alone.

## CXCL10 as a biomarker for dsRNA pathology
Specific biomarkers for neuroinflammatory-based neurodegenerative disease subtypes associated with dsRNA-induced pathology would help identify patients who could benefit from treatments modulating TYK2. We screened potential biomarkers in a genome-wide association study (GWAS) using proteomics data from the UK Biobank[57] and Icelandic DeCODE database[58]. Even though processed on different proteomics platforms (UK Biobank: Olink, DeCODE: SomaScan), we consistently found that individuals with an established partial loss-of-function polymorphism in *TYK2* (rs34536443) in both cohorts have significantly lower plasma CXCL10 levels (UK Biobank: Effect: −0.12, SE = 0.02, P = 2.72 × 10$^{-11}$; DeCODE: Effect: −0.10, SE = 0.02, P = 5.04 × 10$^{-11}$) (Fig. 7a). This minor allele coding mutation (P1104A) leads to a near-complete (~80%) loss of TYK2 function in homozygotes, while heterozygotes have <40% reduction of function[59–61], and its functional effect mimics the action of deucravacitinib[56].

It has been shown that CXCL10 is elevated in patients with AD[62–66]. To study the chemokine CXCL10 as a potential biomarker for cdsRNA-related pathology further, we examined its levels in media from our ReN VM cell-based model system by conducting an electro-chemiluminescence (ECL) immunoassay (Fig. 7b). As expected from our proteomics data showing increased expression of CXCL10 in neural cells transfected with poly(I:C) compared to control (Fig. 6c), we saw significantly increased levels of CXCL10 in media of cells treated with this dsRNA mimetic. The application of baricitinib, ruxolitinib, and deucravacitinib diminished CXCL10 expression to levels comparable to the lipofectamine-only control condition in a dose-dependent manner (P < 0.0001 in all conditions).

Next, to evaluate if this candidate biomarker can be seen in association with TDP-43 pathology, we modified our cell-based assay by treating differentiated ReN VM cells with 1 μM MG-132, a proteasome inhibitor known to induce cytoplasmic aggregation of TDP-43[67,68]. We confirmed translocation of TDP-43 from the nucleus to the cytoplasm by immunofluorescent staining with two different antibodies (Supplementary Fig. 9a,b) and quantified CXCL10 levels in the media by ECL immunoassay. CXCL10 levels were significantly elevated after MG-132 treatment (P < 0.005) and were reduced to control levels by 10 μM deucravacitinib (P < 0.005) but not by baricitinib and ruxolitinib treatment, indicating that selective TYK2 inhibition is most effective in reducing IFN-I mediated CXCL10 levels among the JAK kinases (Fig. 7c).

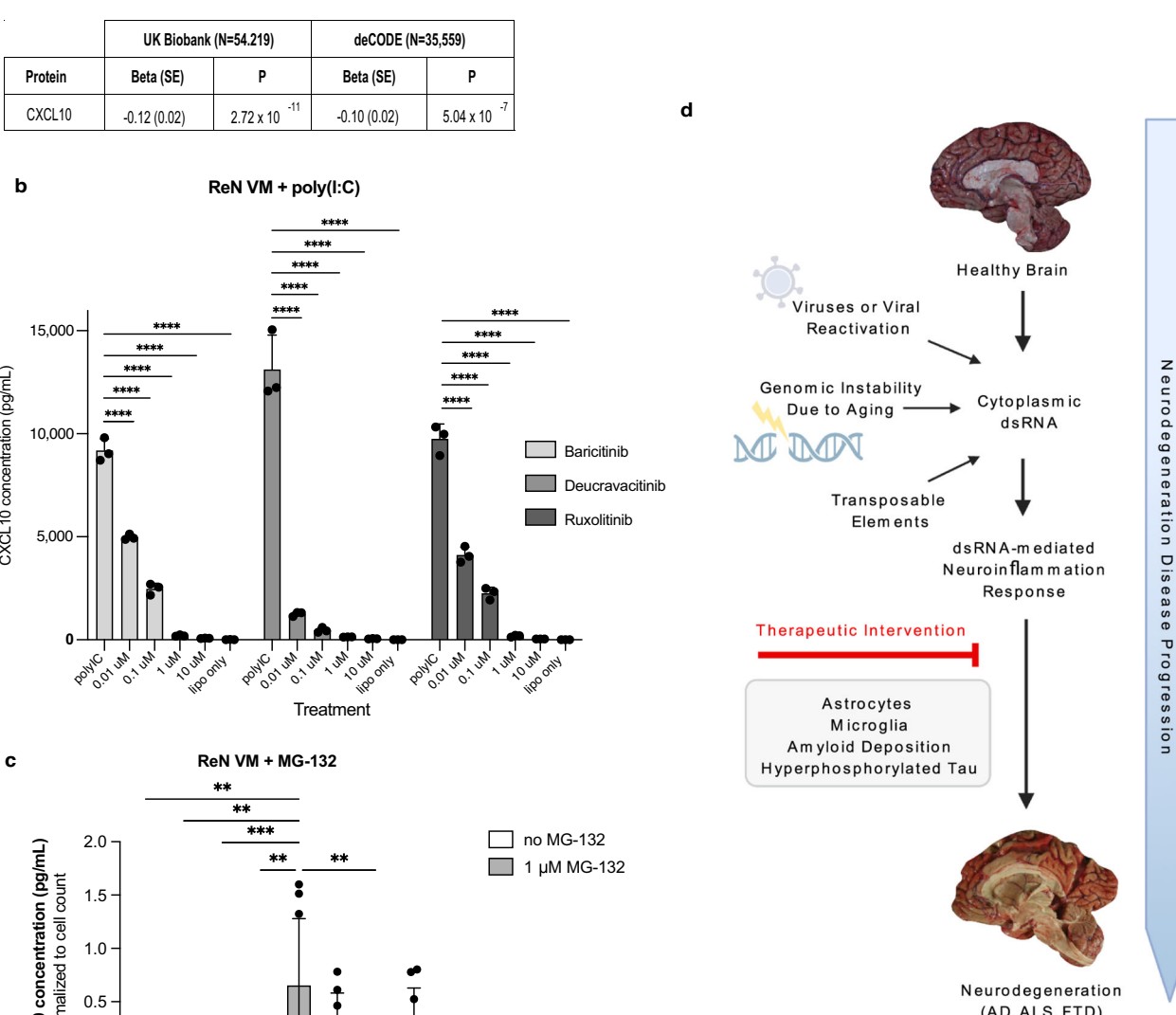

| Protein | UK Biobank (N=54.219) | | deCODE (N=35,559) | |
|---|---|---|---|---|
| | Beta (SE) | P | Beta (SE) | P |
| CXCL10 | -0.12 (0.02) | $2.72 \times 10^{-11}$ | -0.10 (0.02) | $5.04 \times 10^{-7}$ |

**Fig. 7 | CXCL10 as a candidate biomarker for cdsRNA-positive AD. a** An established partial loss-of-function single nucleotide polymorphism (SNP) in *TYK2* (rs34536443) affects plasma levels of CXCL10 based on data from the UK Biobank (Olink proteomics platform)[57] and Icelandic DeCODE database (SomaScan proteomics platform)[58] using genome-wide association tests. Statistical analyses were performed using a two-tailed Z-test. No adjustments were made for multiple comparisons. **b** CXCL10 concentration in pg/ml in the media of ReN VM-derived neural cells treated with different concentrations of baricitinib, ruxolitinib, or deucravacitinib and transfected with poly(I:C) (*n* = 3 per condition; *P* < 0.0001 in all conditions). **c** CXCL10 concentration in the media of ReN VM-derived neural cells treated with 1 μM MG-132 for the translocation of TDP-43 from the nucleus to the cytoplasm without any additional drug (ctrl), or with additional 10 μM baricitinib, ruxolitinib, or deucravacitinib treatment. ROUT (Q = 5%) and Grubbs (a = 0.05)

outlier tests were carried out confirming no outliers (*n* = 9 per condition; no MG-132 ctrl vs. 1 μM MG-132 ctrl *P* = 0.0038, baricitinib without MG-132 vs. 1 μM MG-132 ctrl *P* = 0.0013, deucravacitinib without MG-132 vs. 1 μM MG-132 ctrl *P* = 0.0008, ruxolitinib without MG-132 vs. 1 μM MG-132 ctrl *P* = 0.0017, 1 μM MG-132 ctrl vs. deucravacitinib with 1 μM MG-132 *P* = 0.0049). **d** Schematic overview of hypothesized pathomechanisms underlying cdsRNA in neurodegenerative diseases that focuses on dsRNA-mediated neuroinflammation. Other potential triggers for neuroinflammation are not depicted here for simplicity's sake. Created in BioRender. Daneshvari, S. (2026) {https://BioRender.com/0fpvo9s}. All replicates are biological replicates. All graph bars represent mean values while the error bars indicate the standard deviation. Statistical tests (**c**, **d**): ordinary one-way ANOVA with correction for multiple comparisons by Dunnett's test. Source Data is provided as a Source Data file.

Taken together, these results nominate CXCL10 as a candidate biomarker for the assessment of dsRNA-mediated neuroinflammation using the cerebrospinal fluid (CSF) of patients with AD and could help determine whether these patients would benefit from therapeutic approaches reducing TYK2 activity.

## Discussion

Mounting evidence implicates inflammatory signaling as an important driver of AD progression, but the root causes of neuroinflammation have been elusive. Here, we demonstrate that cdsRNA, a DAMP that triggers IFN-I signaling in neural cells, is spatially coincident with

cytoplasmic pTDP-43 inclusions in brains of patients with AD, and we observe the upregulation of many ISGs in AD brains, consistent with the predicted IFN-I response to cdsRNA. Moreover, we showed elevated levels of cdsRNA in iPSCs with a pathogenic TDP-43 mutation[33] that also shows increased interferon signaling[34], replicating our findings in humans in a regulated cellular model. We improved our machine learning pipeline DRIAD-SP[23] to stratify the assessed cases by expression levels of CEs as a proxy of TDP-43 pathology, as growing evidence indicates the presence of CE-containing mRNA and de novo proteins in AD brains with TDP-43 pathology[24–29]. Using this new DRIAD-SP pipeline, we identified baricitinib and ruxolitinib as hits in a CE-positive subset of AD, further refining the hypothesis of JAK inhibitors as drug candidates to a subset of AD patients with neuroinflammation. Both our CRISPR screen and systems pharmacology analyses implicated the selective inhibition of TYK2 as a target for rescuing cdsRNA-induced toxicity. Our findings are consistent with previous studies (including our own[23]) linking TYK2 to AD[69–72]. Finally, we propose CXCL10 as a candidate inflammatory biomarker in neurodegenerative diseases with cdsRNA-pathology to help identify patients who could benefit from therapies mediating TYK2 activity.

Increased expression of dsRNA from prolonged 3'UTRs in neurons may have a functional role[73], but the accumulation of cdsRNA in neurons may cross a threshold to invoke an additional pathogenic mechanism in AD by serving as a root cause of neuroinflammatory cascades that are triggered within neurons[22,74]. Introducing dsRNA is sufficient to trigger neuroinflammation and neuronal death in the Nd1 and Nd2 mouse models[22], and cdsRNA also induces an innate immune response and death in a differentiated human neural cell-based model that lacks microglia. The mechanism of cell death linked to dsRNA is TYK2-dependent, and IFN-I signaling alone (although it activates TYK2) is not sufficient to trigger cell death, indicating that dsRNA has broader effects than just triggering interferon, such as the modulation of autophagy[75] - perhaps acting through TBK1. However, growing evidence supports microglia activation playing an important role in AD progression, and cdsRNA-mediated neuroinflammation and neural cell toxicity can trigger microglial activation[22,76,77] and consequent propagation of AD pathology[78,79]. Together, "neuroninflammation", an emerging concept which includes endogenous accumulation of cdsRNA from genomic[80,81], mitochondrial[82], and viral[83] sources as well as other DAMPs[34], likely interacts with astrocytic[84] and microglial[85] cascades to drive disease progression.

In a tau-based mouse model, mitochondrial DNA leakage leading to cytoplasmic DNA and STING activation in microglia could contribute to an IFN-I pathogenic mechanism[86]. Additionally, cytoplasmic DNA within neurons could activate this cGAS-STING pathway[34,87], which activates the kinase TBK1, a common node of dsRNA- and dsDNA-mediated innate immune signaling networks. Neither ReN VM nor CX cells show expression of STING. Thus, our results in these cell lines are independent of any cGAS-STING signaling and are specific to the cdsRNA-MDA5/RIG-I axis of IFN-I signaling. The convergence of cGAS-STING and cdsRNA-mediated IFN-I-induced toxicity as co-acting pathologies underlying neurodegenerative diseases may be present in the same patient, suggesting that therapeutic strategies downstream of the convergence of these networks may be effective for both stimuli.

Drug repurposing accelerates testing therapeutic hypotheses expeditiously in clinical trials, and FDA-approved drugs can serve as chemical biology probes to elucidate novel targets. By elucidating the mechanism of action of baricitinib and ruxolitinib, which target the JAK kinase family, which lies downstream of TBK1 signaling, in human neural cells, we identified TYK2 as a target to rescue cdsRNA-induced toxicity. Deucravacitinib reverses dsRNA-induced toxicity at lower doses than either ruxolitinib or baricitinib. Its high selectivity for TYK2 could achieve improved safety profiles relative to baricitinib and ruxolitinib. The inhibition of JAK1-3 is associated with adverse infectious,

embolic, and thrombotic, neoplastic, and gastrointestinal perforation events[88], which have not been reported in people with *TYK2* polymorphisms[59,89] or those treated with a potent TYK2 inhibitor[90]. However, deucravacitinib is unlikely to cross the blood-brain barrier based on its polar molecular structure[55], making it imperative to develop novel brain-penetrant TYK2-selective inhibitors.

Translation of these results will require identification of validated predictive biomarkers of drug action on the inflammatory response. We have identified CXCL10 as a candidate biomarker to detect the activation of cdsRNA-mediated neurodegeneration in the CSF of patients, which affords the possibility of precision medicine in clinical trials for AD and other neurodegenerative diseases. CXCL10 can also lead to recruitment of CD8 + T cells that can further aggravate neurodegeneration[91,92]. In addition to our proposed biomarker, recent studies show that peptides expressed from RNA containing CEs as a result of nuclear TDP-43-depletion can be detected in the CSF of ALS patients[29,93,94] and could thus also be used as a biomarker in AD and other cdsRNA/TDP-43-positive neurodegenerative diseases, such as chronic traumatic encephalopathy[95].

Limitations to our study include examination of only a few brain regions, both in our histologic and our computational analyses. Future studies will analyze more brain regions in cohorts of more diverse patients and at different stages of the disease. This limitation may account for why we do not observe spatially coincident accumulation of cdsRNA foci and NFTs. Recent work illustrating the connection between tau pathology and transposon derepression[96] and one reason for our inability to see this colocalization could be that the toxicity of spatially coincident cdsRNA and NFTs in neurons in the examined brain regions is too high to be able to observe cells expressing both pathologic lesions in autopsy samples. Interestingly, a recent study established a link between TYK2 and the stabilization of tau and subsequent NFT formation, which also establishes TYK2 inhibition as a promising therapeutic approach in tau-mediated pathology[72]. It is possible that both TDP-43 and tau-mediated pathomechanisms can be targeted by TYK2.

Also remaining to be determined is the source of cdsRNA in AD and what other drivers besides TDP-43 could be. While neurons seem to have generally high intrinsic levels of immunogenic dsRNA[73], we hypothesize additional sources of pathogenic cdsRNA (Fig. 7d), including derepression of transposable elements[97] such as inverted Alu elements[19,98], genomic instability that occurs naturally in an aging brain, and the reactivation of viruses[99] such as HSV[100]. In this work, we observed cdsRNA in the presence of TDP-43 pathology in AD brains, yet we do not claim that this is the only setting in which cdsRNA is present in brains with neuroinflammation, and believe that multiple routes for cdsRNA accumulation exist. Future studies will deploy computational methods (quantitative trait locus analysis) and the isolation and sequencing of cdsRNA from autopsied brains from sporadic AD cases.

Finally, since there are no suitable mouse models for AD or similar neurodegenerative processes, including pTDP-43 pathology, there are no other murine models to test our hypothesis other than our Nd1 and Nd2 mice[22]. However, we have generated evidence supporting our findings in numerous human studies, including human postmortem material, RNA-seq, and proteomics data derived from AD patients.

In summary, we hypothesize that cdsRNA-induced innate immune responses contribute to disease progression in the subset of AD patients with co-existing TDP-43 pathology and also in other neurodegenerative diseases, including ALS and FTD (Fig. 7d). Pathogenic cdsRNA may accumulate over the years until it reaches a triggering point as a DAMP by activating PRRs that induce IFN-I signaling and neuroinflammation. This inflammatory pathology occurs in the context of amyloid deposition and hyperphosphorylation of tau, either as a parallel pathomechanism or placed upstream or downstream of known proteinopathies. The involvement of astrocytes and microglia,

as well as cGAS-STING activation in dsRNA-induced toxicity, has been seen in our mouse models[22] and others[34,77]. Finally, therapeutic interventions that target TYK2 to inhibit cdsRNA-induced neuroinflammation may have potential not only in AD but also in other neurodegenerative diseases with TDP-43 pathology, a challenge that will be tackled by personalized medicine approaches in neurodegenerative diseases.

## Methods

This study complies with all relevant ethical regulations and was approved by the MGB IRB (2006P002104). Details can be found within each respective method section.

### Histological staining and imaging

All human tissues were acquired from the MGH MADRC brain bank in accordance with protocols approved by the MGB Institutional Review Board. Informed consent, including consent to publish three or more indirect identifiers, was given by all participants. Participants were not compensated for their contribution to this study. Detailed patient data can be found in Supplementary Table 1. Sex was self-reported by participants. While an effort to achieve balanced representation was made, sex was not directly considered in the study design and data analysis to ensure that findings apply to any sex. Similar to what was previously reported[22], ~5 μm thick formalin-fixed paraffin-embedded (FFPE) sections of the amygdala were dewaxed, followed by antigen retrieval in a Leica Bond Fully Automated Slide Stainer. After deparaffination and rehydration by sequential incubation in Leica dewing solution (3 × 5 min), 100% ethanol (2 × 5 min), 90% ethanol (2 × 5 min), 70% ethanol (2 × 5 min), and distilled water (3 × 5 min), antigen retrieval was achieved with citrate-based antigen retrieval solution (ER2) and steam cooking for 1 hour. Afterwards, sections were cooled to room temperature and rinsed in water and 1× PBS. For permeabilization, 0.03% hydrogen peroxide solution was applied for 8 min. After blocking in tris-buffered saline (TBS)–based Odyssey blocking buffer (cat. # 927-60001, LI-COR, NE, USA) for 1 hour, for immunohistochemistry, primary antibodies (J2 for dsRNA (1:200, cat. # 76651, Cell Signaling Technology, MA, USA); TDP-43 (1:200, cat. #ab193842, abcam, MA, USA); pTDP-43 S409/410 (1:200, cat. # CAC-TIP-PTD-P02, Cosmo Bio, CA, USA); tau (1:100, cat. # NBP2-25162, Novus Biologicals, CO, USA); Aß (1:100, cat. # CST-D54D2, Cell Signaling Technology); pPKR (1:100, cat. # bs-3336R-Cy3, Bioss Antibodies, MA, USA)) diluted in blocking buffer were applied overnight at 4 °C in a humidified chamber. Following a wash with PBS (3 × 5 min), secondary antibodies were applied using the VECTASTAIN ABC Staining Kit (cat. # PK-6200, Vector Laboratories, CA, USA) and DAB Staining Kit (cat. # SK-4100, Vector Laboratories). For Luxol fast blue-hematoxylin and eosin (LHE) staining, the protocol of the Luxol Fast Blue Stain Kit (cat. # ab150675, Abcam) and the Sakura Prisma H&E Stain Kit (cat. # 76318-76, VWR, PA, USA) were followed. For tissue-based cyclic immunofluorescence (CyCIF)[31], unspecific binding sites were blocked, slides were incubated in 3% H₂O₂ and 20 mM HCl in PBS at room temperature with light-emitting diode illumination for 1 hour to reduce autofluorescence, and the first primary antibody set (J2, TDP-43, and pTDP-43 S409/410) was applied. After washing, the secondary antibodies (anti-rabbit Alexa Flour 555 (cat. # A-31572, Thermo Fisher, MA, USA); anti-mouse Alexa Flour 647 (cat. # A-21237, Thermo Fisher)) 1:1000 diluted in blocking buffer were applied for 1 hour in a humidified chamber at room temperature. After a wash, Sytox Blue Nucleic Acid Stain (cat. # S11348, Thermo Fisher) was applied 1:15,000 in blocking buffer for 15 min at room temperature. Following a final washing step, slides were imaged. After this first cycle, the fluorophores were inactivated by 4.5% H₂O₂ and 24 mM NaOH in PBS at room temperature under light exposure for 1 hour. After rinsing, the second primary antibody set (pPKR, tau, and Aß) was applied, and staining was continued in the same manner as described above. Images were acquired with the CyteFinder slide scanning fluorescence microscope (RareCyte Inc., WA, USA) using the Imager5 software (v. 2018, RareCyte Inc.).

### Differential gene expression analysis of brain regions

This Differential Gene Expression protocol was established based on a previously established protocol[101]. In brief, clinical diagnosis was used to define AD versus control conditions within the bulk RNA-sequencing data taken from the ROSMAP[32] and MSBB[35] databases (data obtained from the AMP-AD Knowledge Portal (doi:10.7303/syn2580853)). The analysis focused on a list of 354 pre-selected interferon-related genes. Of these genes, 309 could be found in the analyzed brain regions (from ROSMAP: dorsolateral prefrontal cortex (DPFC), from MSBB: frontal pole (FP), inferior frontal gyrus (IFG), parahippocampal gyrus (PHG), superior temporal gyrus (STG)). 139 genes were significantly upregulated (adjusted P value < 0.05) while 51 were downregulated. The full results with LFC, FC, CI, P values, and adjusted P values can be found in Supplementary Data 1.

### Induced pluripotent stem cell differentiation and treatment

An isogenic pair of a recently established induced pluripotent stem cell (iPSC) line that harbors a pathogenic TDP-43 mutation (TDP-43[+/G298S]) in the FA0000011 genetic background[33,102] was differentiated into NGN2 cortical-like neurons using a recently published protocol[34,103] in a 384-well plate format (10,000 cells per well). Briefly, NGN2 neurons were generated by nucleofecting iPSCs with PiggyBac plasmids[104–106] containing Tet-inducible transcription factors NGN2 and transposase[107]. After 24 hours of culture in media supplemented with CET, the cells were selected based on the presence of nuclear blue fluorescent protein 2 (BFP2) signal and using puromycin (10 μg/mL; cat. # ant-pr-1, InvivoGen, CA, USA), indicating successful plasmid integration. iPSCs were plated on Matrigel-coated (cat. # 354320, Corning) plates and differentiated into post-mitotic neurons using induction media. After three days of differentiation, the cells were dissociated, frozen, and later plated for experiments, where they were cultured with neurotrophic factors and long-term media changes up to day 30 in culture. Where indicated, wildtype FA0000011 cells were transfected with poly(I:C) HMW (cat. # tlrl-pic, Invitrogen, CA, USA) by lipofection with lipofectamine 2000 (dilution factor 3:1000) cat. # 11668027, Thermo Fisher) (final dsRNA concentration: 4-8 μg/ml, depending on poly(I:C) batch effects).

### ReN VM and CX cell culturing and differentiation

ReN VM (cat. # SCC008, Millipore, MA, USA) and CX cells (cat. # SCC007, Millipore) were cultured in ReNcell Maintenance media (cat. # SCM005, Millipore) containing 1:100 of Penicillin-Streptomycin (cat. #30-002-CI, Corning, NY, USA), 20 ng/ml of epidermal growth factor (cat. # GF001, Millipore) and 20 ng/ml of basic fibroblast growth factor (cat. # 03-002, Stemgent, MD, USA) and differentiated into neural cells for one week by removing the growth factors as previously described[22,43], with the difference that the differentiation took place in a separate dish (Extended Fig. 7) to minimize variability downstream.

### TYK2 knockdown in ReN VM cells

A lentiviral vector used to express shRNA for TYK2 (target sequence: TGCAAGCCTGATGCTATATTT), pLV[shRNA]-Puro-U6 > hTYK2[shRNA#1], and the control lentiviral vector used to express scramble shRNA (sequence: CCTAAGGTTAAGTCGCCC TCG), pLV[shRNA]-Puro-U6>Scramble_shRNA, were constructed and packaged by VectorBuilder Inc. (IL, USA). The vector IDs are VB250103-1179zud (TYK2 shRNA) and VB010000-9541pqu (scramble shRNA), which can be used to retrieve detailed information about the vector on vectorbuilder.com. Briefly, ReN VM cells were treated with the virus at a multiplicity of infection (MOI) of 1. After a week-long selection with puromycin at a concentration of 1 μg/ml, knockdown was confirmed by Western blot.

## SH-SY5Y culturing

SH-SY5Y Neuroblastoma cells (cat. # CRL-2266, ATCC, VA, USA) were cultured in 1:1 Eagle's Minimum Essential Medium (EMEM) (cat. # 10-009-CV, Corning) and F12 medium (cat. # 11765-047, Thermo Fisher) containing 1:100 of Penicillin-Streptomycin (cat. #30-002-CI, Corning) and 10% fetal bovine serum (FBS; cat. # 10438-026, Thermo Fisher). Prior to any treatments, cell media was changed to media containing 1% FBS instead of 10% to slow down cell division.

## ReN VM, CX and SH-SY5Y cell treatments

SH-SY5Y cells as well as differentiated ReN VM and CX cells were seeded into 96-well plates (VM/CX: 20,000 cells per well with Matrigel-coating (cat. # 354320, Corning), SH-SY5Y cells: 10,000 cells per well without coating) and where applicable, respective drugs dissolved in dimethyl sulfoxide (DMSO) (baricitinib: HMS LINCS ID # 10354, ruxolitinib: HMS LINCS ID # 10138, deucravacitinib: cat. # HY-117287, MedChem Express, NJ, USA, MG-132: cat. # HY-13259, MedChem Express, etoposide: cat. # HY-13629, MedChem Express, menadione: cat. # M9429-25G, Sigma-Aldrich, MO, USA) were added using a D300e Digital Dispenser (Hewlett-Packard, CA, USA). After 1 hour of incubation at 37 °C, cells were transfected with poly(I:C) HMW where applicable (cat. # tlrl-pic, Invitrogen) by lipofection with lipofectamine 2000 (dilution factor 3:1000, cat. # 11668027, Thermo Fisher) (final dsRNA concentration: 4-8 µg/ml, depending on poly(I:C) batch effects). To test the relevance of IFNAR2, a neutralizing anti-IFNAR2 antibody (cat. # 21385-1, PBL Assay Science, NJ, USA) was added in the respective amount to each well after lipofection and without prior drug treatment. To test whether interferon α and poly(I:C) have comparable effects on neurons, interferon α (cat. # 407294-5MU, Millipore) was diluted in media and added to the cells without prior drug treatment. Western blot samples and cell media for biomarker testing were collected 24 hours (48 hours for MG-132 treatment) post-treatment (before we observe any cell death). Cell viability was assessed 72 hours (VM, SH-SY5Y) and 7 days (CX) post-treatment using the CellTiter-Glo Assay (cat. # G7572, Promega, WI, USA) following the manufacturer's instructions. All conditions were normalized to 0.1% DMSO content. Where applicable, lipofectamine 2000 (VM, CX) or lipofectamine 2000 (dilution factor 3:1000) + 10 µM drug (SH-SY5Y) was used as the respective control. Luminescence was measured on a BioTek Synergy H1 Plate Reader (BioTek, VT, USA) using the Gen5 software (v.3.2). For mass spectrometry, two million ReN VM cells were expended in a 15-cm-dish (2 dishes per replicate) for 4 days and subsequently differentiated for 1 week, then treated with 10 µM of drug, transfected with poly(I:C) after 1 hour of incubation at 37 °C, and 24 hours later, washed with ice-cold 1x PBS and collected by scraping followed by centrifugation at 500 g for 5 min. For the testing of common stressors, time-lapse images were captured every 3 hours for 33 hours using GE In-Cell 6000 Analyzer (GE HealthCare, IL, USA) with the manufacturer's software (v.7.2) under the confocal mode. Each image's total cell surface area and average fluorescence intensity were quantified using Fiji/ImageJ-based plugin (National Institutes of Health, v.1.53c)[108] and plotted using GraphPad Prism (v.9.3.1). Four biological repeats were taken for each group.

## Cell staining and imaging

Differentiated ReN VM cells treated with 1 µM MG-132 were fixed 48 hours post-treatment, and iPSC-derived NGN2 cortical-like neurons were fixed after ten days of differentiation using 4% paraformaldehyde (cat. # 28908, Thermo Fisher) at room temperature and protected from light. Additionally, after three washes with 1 x PBS, ReN VM cells were permeabilized using ice-cold methanol for 10 min. Cells were washed gently with 1 x PBS three times and then incubated with blocking solution (cat. # 927-60001, LI-COR) for 1 hour at room temperature. All blocking solution was removed, and cells were incubated with fresh blocking solution and primary antibodies overnight on a

shaker at 4 °C. Primary antibodies used were anti-TUJ1 (1:250, cat. # TUJ, Aves Lab, CA, USA), K1 anti-dsRNA (1:250, cat. # 10020200, Scicons, Susteren, The Netherlands), and anti-TDP-43 (1:100, cat. # 3448, Cell Signaling Technology, and 1:500, cat. #10782-2-AP, Proteintech, IL, USA). For poly(I:C) antibody incubation controls, poly(I:C) was incubated with the dsRNA primary antibodies at a ratio of 5x poly(I:C): 1x dsRNA primary antibody for 1 hour before being added to cells. Three PBS washes were done, and cells were incubated with secondary antibodies based on species at a 1:500 (iPSCs) and 1:1000 (ReN VM) dilution in the blocking solution for 1 hour at room temperature on a shaker. Hoechst 33342 (cat. # H3570, Thermo Fisher) was added at a 1:1000 dilution to the well with iPSCs for the final ten minutes. For ReN VM cells, cells were first washed three times with 1 x PBS, and then Hoechst was added to the wells in a 1:5000 dilution in PBS for 30 min at room temperature. Secondaries used were donkey anti-chicken Alexa Fluor 488 (cat. # 703-545-155, Jackson ImmunoResearch, PA, USA), donkey anti-rabbit Alexa Fluor 568 (cat. # A10042, Thermo Fisher), donkey anti-mouse Alexa Fluor 647 (cat. # A31571, Thermo Fisher), and donkey anti-rabbit Alexa Fluor 488 (cat. # A21206, Thermo Fisher). After three final washes with 1 x PBS, confocal images were acquired with an Image X-Press Micro Confocal (Molecular Devices, CA, USA) using MetaXPress (v.6.5.3.427). Images were taken with a 20x (ReN VM) or 40x (iPSC) objective with four z-stacks (1.5 uM step size) per field for between 25 and 55 fields per well. All groups were imaged with the same acquisition settings within each experiment. Automatic quantifications were performed using a custom Fiji/ImageJ-based plugin (v.1.53c)[108] that measures integrated intensity of K1 dsRNA antibody within TUJ1-positive cells.

## DRIAD-SP: RNA-sequencing data processing and TDP-43 status prediction

RNA-sequencing reads of brain specimens from AD patients in the ROSMAP study, along with their corresponding clinical annotations, were downloaded from the AMP-AD Synapse portal at https://adknowledgeportal.synapse.org. RNA-sequencing reads from ALS patient neurons that were sorted for the presence or absence of TDP-43 were downloaded from Sequence Read Archive (SRA) accession GSE126542[40]. We quantified transcripts using Salmon (v1.9.0)[109] against release 107 of the hg38 human transcriptome from Ensembl. We amended the transcriptome with three transcripts that have not yet been annotated in Ensembl, but that are known to be associated with loss of nuclear TDP-43[14]. Specifically, we included two splice variants of UNC13A, each including an additional CE between the canonical exons 20 and 21 (hg38; chr19: 17,642,414–17,642,541 (CE1); chr19: 17,642,414–17,642,591 (CE2)), and one splice variant of STMN2, which includes an alternative exon 2 termed exon 2a (hg38; chr8:79,616,822–79,617,048).

Given our quantification of TDP-43 pathology-associated transcripts, we classified patient samples as TDP-43 pathology-positive or -negative based on the number of expressed CEs. We dichotomized transcript abundance using thresholds set above the lowest non-zero peak in the abundance histograms (TPM > 1.3 STMN2 short; TPM > 0.09 for UNC13A-CE1; TPM > 0.08 UNC13-CE2). This threshold determines whether each individual CE is present or absent. Next, patient samples expressing one or none of the three CEs (STMN short, UNC13A-CE1, or UNC13A-CE2) were considered TDP-43 pathology-negative, and patient samples expressing two or all three of the CEs were considered TDP-43 pathology-positive. Using this classification, the proportion of TDP-43 pathology-positive samples is approximately 37%. Our classification is a conservative prediction, given that the clinical incidence of TDP-43 inclusions in AD patients is up to 57%[10]. To assess if ISGs are upregulated in cases that we deemed TDP-43 pathology-positive, a list of ISGs was curated based on Supplementary Tables 2–10 in Schoggins et al.[110] and the expression of these genes was compared to the expression in TDP-43 pathology-negative cases.

## DRIAD-SP: Disease stage prediction and gene set significance assessment

For assessing how well any given gene set predicts AD progression, we used an updated variant of the published DRIAD-SP methodology[23]. For predicting disease stage, we collapsed transcript abundances to gene-level abundances using tximport (v.1.43.0)[111] and msigdbr (v.24.1). The entire transcriptional feature space was filtered down to about ~20k protein-coding genes in the human genome to ensure that only genes with gene products that can conceivably be targeted using conventional small molecule inhibitors are included. Every AMP-AD specimen was assigned a label of disease severity based on the following mapping to the Braak annotations: A−early (Braak 1−2), B−intermediate (Braak 3−4), and C−late (Braak 5−6). We set up an ordinal regression task classifying patients according to their disease stage using the ordinalRidge R package (v.0.1; https://github.com/labsyspharm/ordinalRidge). We used the gene expression measurements from AMP-AD as predictors, filtered to only contain genes in the current gene set under investigation. This way, the performance (AUC) of a model utilizing a given gene set can be viewed as a measure of how strongly the given genes are predictive of AD progression. To address overfitting, ordinalRidge utilizes a ridge regularization term that penalizes the L2-norm of feature weights. No LASSO regularization was used, as it induces sparsity and excludes features that were specifically preselected to be included in the model.

Model performance was evaluated through leave-pairs-out cross-validation. For a given regression task, each example in the dataset was associated with an example from the other classes that was the closest match in age. If there were multiple candidates for the age match, the pairing was selected uniformly at random. The resulting set of age-matched pairs was evaluated in a standard cross-validation setting, by asking whether the later-stage example in each withheld pair was correctly assigned a higher score by the corresponding predictor. The fraction of correctly ranked pairs constitutes an estimate of the area under the ROC curve[112].

Comparing model performances across gene sets directly is not possible, as their set sizes can differ. Therefore, we normalized AUC values by relating them to performances observed on size-matched random gene sets. For each gene set of interest, 1000 random gene sets of matching sizes were sampled from a uniform distribution over the protein-coding space. After evaluating all lists through cross-validation, an empirical P value was computed as the fraction of background sets that yield higher predictor performance than the gene list of interest.

## CRISPR screen

Low passage (P2) ReNcell VM cells stably expressing Cas9 were expanded and transduced with lentiviral particles containing the Brunello library[47]. Lentiviral transduction efficiency was tested by plating cells into two 6-well plates and varying volumes of stock preparation of Brunello library lentiviral vectors with 8 µg/ml of polybrene. One plate was treated with puromycin, the other plate remained untreated, and the proportions of cells measured for each condition using CellTiter-Glo were used to calculate the efficiency, which was found to be at 84% averaged from triplicates. Transduced cells were expanded in the presence of puromycin until the required number of cells was achieved to obtain triplicates of 1000 cells per sgRNA. Then, cells were differentiated as described above. After seven days, 3 µg/ml poly(I:C) in complex with lipofectamine or lipofectamine alone (dilution factor 3:1000) as a control was added to each flask. After 3 days of treatment, genomic DNA was isolated from each group of cells. PCR and sequencing were performed according to standard protocols[113,114]. For analysis, the algorithm STARS was used to determine P-values[113], the read counts were normalized to reads per million, and then log2 transformed, and the LFC of treatment and control were compared to the lipofectamine controls. Genes were considered a hit if the LFC was equal to or greater than 2.0 after rounding to the first decimal place.

## Pathway enrichment analysis

Pathway enrichment analysis of genes that rescued neural cell death was performed by selecting the 100 genes with the most significant sgRNA enrichment and querying the "Reactome_Pathways_2024" database[51] using the R (v.4.4.2) package enrichR[48–50] (v.3.4). Enrichment analysis of differentially expressed genes in AD samples was performed using the fgsea R package[115] (v.1.32.4) with KEGG Medicus gene sets[36]. Genes were ranked using the metric -sign(log2FoldChange) × log10($p$-value).

## Mass spectrometry

First, cells were lysed in lysis buffer (2% SDS, 150 mM NaCl, 50 mM Tris pH 8.5) supplemented with protease and phosphatase inhibitor (cat. # 11873580001 and 4906845001, Millipore) and homogenized with a QIAshredder column (cat. # 79656, Qiagen, Hilden, Germany). Lysates were reduced with freshly prepared dithiothreitol (DTT) (final concentration: 5 mM) and heated at 37 °C for 1 hour. Samples were then alkylated with iodoacetamide (final concentration: 20 mM) for 25 min with 50 mM ammonium bicarbonate in the dark whereupon the reaction was stopped by adding DTT (final concentration: 50 mM). After methanol/chloroform precipitation, the purified protein was solubilized in freshly prepared 8 M urea in 200 mM EPPS, at pH 8.5 for 30 mins at 37 °C. For digestion, the urea concentration was diluted with 200 mM EPPS to 4 M final concentration and 2% acetonitrile (v/v) was added. Digestion was performed using lysyl endopeptidase® Lys-C (2 mg/ml; cat. # 121-05063; Wako, Osaka, Japan), (enzyme-to-substrate ratio: 1:50) for 3 hours at 37 °C. Following dilution of the sample with 200 mM EPPS to a final urea concentration of 1.6 M, further digestion was performed by the addition of trypsin (cat. # V5111, Promega), (enzyme-to-substrate ratio: 1:100) for 8 hours at 37 °C. A small aliquot was analyzed for its missed cleavage rate by mass spectrometry. Equal amounts of digested protein (60 µg) were removed from each sample and labeled using TMTpro 16plex Mass Tag Labelling Kit (TMT; cat. # A44520, Thermo Fisher) following the manufacturer's instructions. Labeling efficiency of > 95% was determined along with ratio checks by mass spectrometry MS³, while the labeling reactions were stored at -80 °C. To quench the reaction, hydroxylamine was added to a final concentration of 0.5% (v/v) for 10 mins. After acidification using formic acid, equal amounts of labelled peptide were pooled from each sample (as judged from ratio check data). Then, the solvent was evaporated. The labeled peptide multiplex was desalted by solid-phase extraction (SPE; SepPak tC18 Vac RC Cartridge). The peptides were fractionated by HPLC alkaline reverse phase chromatography (Agilent 1200 Series) into 96 fractions and combined into 24 samples. Lastly, peptides were desalted over Stage Tips[116] prior to MS analysis.

A MultiNotch SPS-MS³ TMT method[117] was used on an Orbitrap Lumos mass spectrometer (Thermo Fisher) operated with Tune (v.3.4) as well as Xcalibur (v.4.5.445.18) and coupled to a Proxeon EASY-nLC 1200 liquid chromatography (LC) system (Thermo Fisher). Samples were injected onto a 40 cm, 100 µm (internal diameter) column packed with 2.6 µm Accucore C₁₈ resin (flow rate of 450 nl/min). Over the course of 4 hours, the peptide fractions were separated through acidic acetonitrile gradients by the LC before being injected into the mass spectrometer. First, an MS¹ spectrum (Orbitrap analysis; resolution 120,00; mass range 400–1400 Th) was taken. Then, an MS² spectrum was collected after collision-induced dissociation (CID, CE = 35) with a maximum ion injection time of 150 ms and an isolation window of 0.7 Da. For TMT quantification of peptides, MS³ precursors were fragmented by high-energy collision-induced dissociation (HCD, CE = 40%) and analyzed in the Orbitrap at a resolution of 50,000 at 200 Th. Further details can be found in a previously published article[118].

A Sequest (v.28rev.12)-based in-house software was used to search peptides against a human database with a target decoy database strategy and a false discovery rate of 1%. Oxidized methionine residues (+15.9949 Da) were dynamically searched, along with static modifications for alkylated cysteines (+57.0215 Da) and the TMTpro reagents (+304.207145 Da) on lysines and the N-termini of peptides. Relative protein quantification required a summed $MS^3$ TMT signal/noise > 200 over all TMT channels per peptide and an isolation specificity > 70% for any given peptide. Quant tables were generated and exported to Excel for further processing with JMP Pro (v.16), GraphPad Prism (v.9.3.1), and monocle (v.1.1). More details on the TMT intensity quantification and certain parameters can be found in another recent publication[119].

## Western blotting

First, 15 μl of Laemmli Sample Buffer (cat. # 1610737; Bio-Rad Laboratories, CA, USA) with 1:20 2-mercaptoethanol was added to each well after aspiration of the media. Subsequently, the lysates were boiled at 100 °C for 5 min. Then, the samples were loaded onto a precast gel (cat. # 4569036; Bio-Rad Laboratories, CA, USA) which was run in 1x running buffer diluted from a 10x Tris/Glycine/SDS stock (cat. # 1610772; Bio-Rad Laboratories) at 160 V for 40 min. The transfer was performed using a PVDF membrane (cat. # IPFL00010; Millipore) and a 1x transfer buffer diluted from a 10x stock (cat. # PI35040; Thermo Fisher, MA, USA) mixed with 20% methanol. For 90 min, the gel was blotted at 4 °C at 90 V. Finally, unspecific binding sites were blocked by phosphate-buffered saline (PBS)-based Odyssey® blocking buffer (cat. # 927-40150; LI-COR) for 1 hour at room temperature. Then, primary antibody (pSTAT1 Y701 (cat. # 9167S; Cell Signaling Technology); beta actin (ACTB, cat. # 3700; Cell Signaling Technology); TYK2 (cat. #ab303500, abcam); STING (cat. #13647, Cell Signaling Technology)) diluted 1:1000 in blocking buffer was applied overnight at 4 °C. Then, the blot was washed with tris-buffered saline (cat. # sc-262305; Santa Cruz, TX, USA) containing 0.05% Tween 20 (cat. # BP337-500; Thermo Fisher) (3 x 5 min). Afterwards, secondary antibody (IRDye 800CW (cat. # 926-32211; LI-COR); IRDye 680RD (cat. # 926-68070; LI-COR)) diluted 1:2500 in blocking buffer was applied for 1 hour at room temperature. After three final washes, the blot was imaged with the Odyssey® DLx Imager (LI-COR). Images were analyzed using ImageStudioLite (v. 5.2.5, LI-COR). Unprocessed scans of all blots are provided in the source data file.

## Genome-wide association tests using a *TYK2* polymorphism

We used an established partial loss-of-function in *TYK2* (rs34536443) to estimate the effect of pharmacological TYK2 inhibition on the circulating CXCL10 concentration[59–61]. Summary genetic association data was obtained from a genome-wide association study (GWAS) using the Olink proteomics platform in 54,219 participants of British ancestry in the UK Biobank[57]. Genome-wide association tests were performed using REGENIE (v.2.2.1) in a two-step procedure to account for population structure in a population of European ancestry (n = 34,557). Protein levels were inverse-rank normalized. Association models included the following covariates: age, $age^2$, sex, age × sex, $age^2$ × sex, batch, UK Biobank center, UK Biobank genetic array, time between blood sampling and measurement, and the first 20 genetic principal components of ancestry. Summary genetic association data on these measures were also obtained from a GWAS of these measures using the SomaScan platform in 35,559 individuals of Icelandic ancestry in DeCODE[58]. Rank-inverse normal transformed proteins were adjusted for age, sex, and sample age. Residuals were re-standardized using rank-inverse normal transformation, and standardized values were used in genome-wide association testing using the linear mixed model implemented in BOLT-LMM (v.2.4.1)[120].

## Electrochemiluminescence (ECL) biomarker assay

To measure CXCL10 (IP-10) biomarker levels in collected cell media, we used the commercially available V-PLEX assay from Meso Scale Diagnostics (MSD), LLC (cat. # K151NND, MSD, Cambridge, MA, USA). We used the proposed procedures with the exception that a 1:100 dilution of cell media was required for the sample values to fall within the 0.37-500 pg/ml dynamic range of the assay. All plates were analyzed under the MESO QuickPlex SQ 120 (Model # 1300) plate reader, and concentrations were quantified via the Meso Scale Discovery WorkBench (v.4.0). Samples with concentrations reported as "NaN" were quantified as zero, as these concentrations fell below the dynamic ranges reported above.

## Statistics & reproducibility

Statistical analyses were conducted on GraphPad Prism (v.9.3.1) for unpaired *t* tests with two tails and ordinary one-way analysis of variance (ANOVA) with post hoc tests unless indicated otherwise in the figure legend. Correction for multiple comparisons was either not conducted or done using Dunnett's statistical hypothesis testing or the Benjamini-Hochberg procedure (see figure legend for details). Normality tests (either Shapiro-Wilk, D'Agostino-Pearson omnibus, Anderson-Darling, or Kolmogorov-Smirnov test with Dallal-Wilkinson-Lilliefor P value where appropriate and applicable based on replicate number) were conducted with a significance level of 0.05 after outlier testing and exclusion (ROUT Q = 1% and Grubbs alpha = 0.05). If there was no difference in significance with and without the respective outliers, the outliers were still included in the figure unless otherwise pointed out in the figure legend to ensure that as few values as possible were excluded for a comprehensive representation of the data. No outliers were eliminated from the dataset unless indicated in the figure legend. In the few cases where normality tests failed, a Wilcoxon Rank Sum test was conducted instead of a t-test, which resulted in the same significance as the t-test. For image quantification, data were plotted and statistically analyzed on R-Studio software (v.4.3.3). For all other plots, ggplot (v.3.4.2), ggbeeswarm (v.0.7.2), and Complex-Heatmap (v.2.22.0) were used. For Western blot quantification, ImageStudioLite (v.5.2.5) was used to quantify signals. Boxplots and bars represent the mean and standard error of the mean, respectively. All error bars indicate the standard deviation. All replicates are biological replicates. Differences were considered significant if $P < 0.05$. P value style: $P > 0.05$ and $P = 0.05$ (no asterisks), $P < 0.05$ (*), $P < 0.01$ (**), $P < 0.001$ (***), $P < 0.0001$ (****). Statistical details for each figure are outlined in the figure legend. Sample sizes were chosen in a way that ensures at least three (except for Fig. 4g which is a proof-of-concept and does not require statistics) and up to 24 biological replicates per condition, which was sufficient as the replicates were enough to achieve robust and significant results. No statistical method was used to predetermine sample size. All attempts at replication were successful. Randomization was not relevant to our study as all grouping was done based on phenotype, differences in treatments, and/or diagnosis. Blinding was implemented in our study during the examination of human brain histological sections. This was the only experiment that required blinding in any way.

## Reporting summary

Further information on research design is available in the Nature Portfolio Reporting Summary linked to this article.

# Data availability

For all compounds with an HMS LINCS ID, the compound information, including the vendors they were purchased from, can be found on the HMS LINCS website (https://lincs.hms.harvard.edu/). The proteomics raw data and search results generated in this study have been deposited in the ProteomeXchange Consortium via the PRIDE[121] partner

repository under accession code PXD043641. A reporting summary for this article is available as Supplementary Information file. Source Data are provided with this paper. The source data underlying Figs. 1c, d, 4d–h, 5, 6a,b, 7b,c, Supplementary Figs. 1c, 2b, 8a–f, h, and 9 are provided in the Source Data file. The bulk RNA-sequencing data used in this study are available in the ROSMAP[32] and MSBB[35] databases obtained from the AMP-AD Knowledge Portal under the accession code syn2580853 on Synapse [https://doi.org/10.7303/syn2580853]. Source data are provided with this paper.

## Code availability

Scripts to fully reproduce the tables and figures represented in this manuscript are provided on GitHub (https://doi.org/10.5281/zenodo.18100691)[122].

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

## Acknowledgements

We thank Marina Avetisyan and Rohan Bhukar for constructive comments, Kyle Evans for technical support, and Brad Hyman for sharing equipment and insightful discussions. We thank R01 AG058063 (awarded to M.W.A. and P.K.S.); R01 AG078297 (awarded to M.W.A. and B.J.W.), U54CA225088 (subaward to M.W.A. of overall award to P.K.S.), QFASTR award from MGH (awarded to M.W.A.), the CART fund (awarded

to M.W.A.), and the MassCATS program (awarded to M.W.A.) for support. M.K. is a CPRIT scholar in Cancer Research (RR220032). I.T., A.D. and J.Y. acknowledge support from the UK Dementia Research Institute at Imperial College London, and NIHR Imperial Biomedical Research Centre (BRC). The results published here are in part based on data obtained from the AMP-AD Knowledge Portal (doi:10.7303/syn2580853). These data were generated from postmortem brain tissue collected through the Mount Sinai VA Medical Center Brain Bank, led by Dr. Eric Schadt from Mount Sinai School of Medicine, and through the Rush Alzheimer's Disease Center, led by Dr. David Bennett from the Rush University Medical Center, Chicago.

## Author contributions

L.E.K., S.R., and M.W.A. designed the experiments. L.E.K., S.R., M.T., Sh.D., A.C., M.A., A.S., Y.S., and G.Z. contributed to the investigative work, data curation, and the validation of the presented findings. C.H. and A.S. implemented the machine learning framework and conducted the drug prediction analysis. Su.D. contributed to the computational analysis of RNA-sequencing data regarding interferon signaling. C.H. conducted all gene enrichment analyses. C.M., S.P., and B.J.W. carried out all experiments on iPSCs. G.B., R.E., and M.K. collected, analyzed, and visualized the proteomics data. F.P. and D.E.R. contributed to the CRISPR screen. J.Y., A.D., and I.T. conducted the GWAS study. L.E.K. and M.W.A. wrote the original draft of the manuscript. L.E.K. and Sh.D. contributed to the visualization of the results. All authors contributed to the figures and manuscript reviewing and editing. P.S. contributed to the conceptualization and supervision of this project. M.W.A. was responsible for supervision, project administration, and funding acquisition.

## Competing interests

M.W.A. is a consultant for Sudo Biosciences Limited, TicTwo Therapeutics, and Transposon Therapeutics. A.S. is an employee at Etiome, a subsidiary of Flagship Pioneering. F.P. is an employee of Merck Research Laboratories. D.E.R. receives research funding from members of the Functional Genomics Consortium (AbbVie, BMS, Jannsen, Merck, Vir) and is a director of Addgene, Inc. The remaining authors declare no competing interests.
