## [Transparent Peer Review file · Nature Communications]

TYK2 mediates neuroinflammation in Alzheimer's disease brains with TDP-43 pathology

Corresponding Author: Dr Mark Albers

Version 0:

Reviewer comments:

Reviewer #1

(Remarks to the Author)

Summary:

The manuscript from König et al, presents an interesting hypothesis which is partly supported by the evidence from multiple models/databases, including their own. The hypothesis suggests that cytoplasmic dsRNA is a proxy for TDP-43 pathology in AD which triggers INFalpha/beta signaling and downstream JAK/STAT pathway, which upon inhibition ameliorate neuroinflammation and cell death in AD cell models. The proposed mechanism is compelling and findings could be provide significant for the field of neurodegeneration. However, the data presented here are mostly correlative rather than causative to infer mechanism. While these data are intriguing, there are several questions that arise.

Major points:

- In fig 1b, authors' claim of not finding neurofibrillary tangles (NFTs) and Abeta plaques in AD brain samples needs to be verified by using specific antibodies/stains that identify aggregated forms of tau and Abeta, such as MC-1 antibody (Tau (MC-1) | ALZFORUM) for NFTs, or stains such as methoxy-X04 (Klunk et al; 2002 PMID: 12230326) for Abeta plaques. Also, the brain areas that were examined with immunostaining are not mentioned.
- In Fig 1c, the intensity of all channels in the representative control panel is much lower than the TDP-43 mutant panel. In the graph, what is the dsRNA intensity normalized to?
- It appears that different brain regions are examined for upregulated ISGs and for CEs. Using the same brain regions would clarify the inference. For example, lines 137-140 state "The parahippocampal gyrus (PHG), a brain region that shows atrophy early on in disease progression^{39,40}, had the greatest number of upregulated ISGs, followed by the dorsolateral prefrontal cortex (DPFC) and the inferior frontal gyrus (IFG)". However, while examining CEs in the same RNA seq database, in lines 187-189, "Consequently, we restricted our analysis to ROSMAP sequencing data from the posterior cingulate cortex (PCC), a brain region impaired early on in disease progression and therefore highly relevant for our study⁴⁵," a different brain region is used. To make useful correlations, it will be helpful to examine these mRNA changes in the same brain areas.
- How do CEs relate to cdsRNA? Are the authors correlating them in any way? How does the statement in lines 204-206, "These findings support our hypothesis that cdsRNA associated with TDP-43 dysfunction contributes to IFN-I-mediated neuroinflammatory processes occurring in AD patients' brains", relates to the findings regarding CEs in the same result subsection. This point should be clarified and discussed further.
- Choice of using Ren VM cell line over iPSC-derived NGN2 neurons for the CRISPR screen should be defended/explained further. It is confusing to understand why the authors would use a "neural" line that displays markers of neurons and glial cells and yet be considered "differentiated". In contrast, using iPSC-derived NGN2 neurons will help decipher a more cell-autonomous signaling mechanism. It will strengthen the paper to validate the main findings in iNeurons as well.
- In fig 3c and 3d, there is more than 50% cell death in polyIC treated cells. With that in mind, in fig 3e, is equal protein loaded for the Western blot? It would help to normalize pSTAT1 levels over total protein and with total STAT1 than with only actin. It would also help to see the entire blot to make a firm assessment of the results presented. The same applies to extended fig 10. These are some examples of the pSTAT1/STAT/actin WB (Boriero et al, 2020 The anti-STAT1 polyphenol myricetin inhibits M1 microglia activation and counteracts neuronal death - Boriero - 2021 - The FEBS Journal - Wiley Online Library; Zhao et al, 2022 STAT1 Contributes to Microglial/Macrophage Inflammation and Neurological Dysfunction in a Mouse Model of Traumatic Brain Injury | Journal of Neuroscience (jneurosci.org))
- In extended fig 11, it will help to see higher magnification of the images to appreciate the translocation along with quantification. Use of standard total TDP-43 antibody (ProteinTech, Cat# 10782 (AA 203–209 and AA near N-terminus/R <https://rdcu.be/dYpSn>)) will also help verify this result.

- Dose-dependent curves for JAK or Tyk2 inhibitors to show gradual changes in pSTAT1 and CXCL10 levels will be more convincing.
 - Adding a pathway schematic describing the working mechanism proposed by the authors (similar to extended fig 2a) in the main figures will help assimilate the findings of the paper derived from different cell lines/ databases. Careful assessment of correlation and causation needs to be made.
 - Overall, the images in the paper can be improved by using higher resolution/magnification (and quantification wherever applicable) as well as use of complementary antibodies such as total TDP-43 (ProteinTech, Cat# 10782 (AA 203–209 and AA near N-terminus/R <https://rdcu.be/dYpSn>), 6E10 antibody (monomeric Abeta and plaques, Biolegend, Cat# 803014) and Tau (HT7) antibody (Thermo Fisher Cat#MN1000).
 - It will be useful for the authors to speculate on cell-autonomous and non-cell-autonomous features of their proposed mechanism in the disease with regard to IFN-I signaling and neuroinflammation. The proposed mechanism in this manuscript could answer important questions regarding neuronal-glia crosstalk in neurodegeneration and is worth exploring further.
- Minor points:
- Line 197, “tau” should be replaced with “TDP-43”

Reviewer #2

(Remarks to the Author)

König et al. aim to understand the role of TDP-43 inclusions in Alzheimer’s Disease, how those inclusions may drive toxicity, and present candidate drugs for reversing that toxicity. Another goal is to develop novel biomarkers for AD using this information.

König et al. present compelling data that link TDP-43 pathology, cytosolic double-stranded RNA, and interferon induction in human patient tissue, iPSC-derived neurons, and a neuron-like cancer cell line. They then train their drug repurposing pipeline using cryptic exon expression to identify candidate drugs for AD with TDP-43 pathology. This pipeline identified a number of tyrosine kinase inhibitors. They show that inhibition of tyrosine kinase signaling can rescue dsRNA-induced cell death by decreasing interferon signaling. Finally, they establish a link between an established TYK2 loss of function variant and decreased CXCL10 levels in human plasma.

There are two major weaknesses in the paper. First, due to reasonable technical limitations, the authors use indirect measures of cdsRNA accumulation and/or TDP-43 pathology and draw conclusions that may be consistent with their hypotheses, but do not rule out others in a satisfactory way. This leads to contradictions in their proposed model between the first and last parts of the text as written. Second, they draw a number of conclusions based on their drug treatments at concentrations that are much higher (in some places >100-fold) than the IC50s reported in the paper. Thus, the specificity of their proposed mechanisms are not fully supported by their data. These two weaknesses can be addressed as follows:

Major Points:

1. The authors stratify patients and controls by pTDP-43 pathology but do not ever tell the reader what the stratification strategy is. This is especially important considering the small number of patients in this study.
2. It is important to show that TDP43 aggregation and/or cytosolic TDP43 and/or phospho-TDP43 is co-incident with dsRNA accumulation in Figures 1A and 1B. Quantitation of this would provide very strong evidence for their proposed model. How many TDP-43 positive cells are dsRNA negative and vice-versa? Does TDP43 aggregation/mislocalization induce dsRNA accumulation in cells adjacent to the cells with aggregation/mislocalization?
3. The authors propose that cdsRNA mediates toxicity at least in part via induction of interferon signaling. However, interferon induction can be due to many factors (as mentioned in lines 142-144 and 395-404 in the text), especially in AD. The connection between TDP43 pathology/cdsRNA accumulation and interferon induction would be much better supported if the data in Extended Data Figure 2 contained cdsRNA staining and a quantification of co-localization and/or intensity of pPKR, cdsRNA, and TDP43 pathology. Similarly, a comparison of transcriptomics (Figure 1D and Extended Data 4) from AD patients with and without TDP-43 pathology would make a much stronger argument in support of their model, rather than comparing AD+TDP43 pathology versus healthy controls.
4. To prove the specificity of their hits, the authors should validate their initial CRISPR-screen phenotype with knockdown or knockout of at least IFNAR2, IRF9, and TYK2. Using these lines, they should then show that treatment with the identified tyrosine kinase inhibitors does not rescue toxicity when the above genes are knocked out or knocked down. Similar experiments should be done with knockdown of JAK2 and JAK1 to show specificity for TYK2. Relatedly, the authors begin treatment with deucravacitinib at 10nM (50-fold higher than the reported TYK2 IC50, but also 10-fold higher than the JAK1 IC50) and see almost complete rescue of the polyIC survival phenotype. Showing a concentration dependence for rescue would provide much stronger evidence for the authors’ model.
5. The model in Figure 4D does not take into account other possible sources of neuroinflammation or the role of TDP-43 pathology, as noted in the text (lines 395-404). The model should be updated so that it does not appear the authors are proposing that only cdsRNA can cause neuroinflammation to cause disease.
6. While the data is sound, it is unclear how relevant ReN VM conditioned media are to human plasma levels of CXCL10. Providing some references that show that conditioned media can be extrapolated to human plasma biomarker levels would make the argument for CXCL10 as an AD biomarker much stronger. To directly compare the mechanisms with Figure 5a, the authors could use gene editing to make the identified SNP and measure CXCL10 levels. Alternatively, directly examining CXCL10 plasma levels for AD versus healthy controls would be sufficient.

Minor Points:

1. For Figure 1C, it would be important to show that TDP43 is mislocalized with TDP43 IF.
2. The heatmap in Extended Data 4 is much more informative than in Figure 1D
3. A sentence or two introducing the RenVM cells would clarify those cells' relevance to the reader.
4. It is unclear how hits are called in the CRISPR screen. Please add this to the methods or in the results section.
5. It would strengthen the authors' argument that cdsRNA toxicity is mediated by interferon if they showed gene enrichment analysis (e.g. using ENRICH: <https://maayanlab.cloud/Enrichr/>) of their screen hits.
6. It would help with clarity to have the drug targets listed with the drugs in Figure 3 and Extended Data 10.
7. The authors mention that cGAS-STING activation can play a role in dsRNA-induced toxicity. If possible (not needed for publication), it would shed a huge amount of light on the relative contribution of the authors' proposed mechanism versus cGAS-STING dsRNA sensing to perform those experiments in a cell line that does not express cGAS-STING.
8. It may be worth discussing a recent paper that identified TYK2 (PMID 39528671) as a regulator of tau levels and turnover in the discussion section.
9. Generally the paper would be improved if more of the extended data were in the main figures.
10. I cannot offer expert feedback on the technical merits of the DRIAD-SP training or framework.

Reviewer #3

(Remarks to the Author)

Summary of the key results

Previous work by the Albers group had identified cytoplasmic double stranded RNA (cdsRNA) in the frontal and motor cortex in post-mortem tissue of ALS-FTD patients with C9ORF72 repeat expansion. CdsRNA was proposed to be produced in this model from antisense sense transcription of RNA from G4C2 expansion. CdsRNA was found to colocalize with phosphorylated TDP-43 in patient samples. (Rodriguez et al 2021 Science Translational medicine). They found cdsRNA to be toxic to neural cells and this toxicity could be rescued with treatment by FDA approved JAK1/2 inhibitors Ruxolitinib and Baricitinib.

Separately the Albers group developed a machine learning model to aid the repurposing of FDA approved drugs for use in Alzheimer's Disease (AD). The model compares gene expression changes at different AD severities with gene perturbations induced in neural cells following treatment with FDA approved drugs. They found the gene list that comprises perturbed genes follow Jak1/2 inhibition by ruxolitinib was able to reliably predict early vs late disease stages in addition to inhibitors for kinases in the ULK and NEK families. (Rodriguez et al. 2021 Nature Communications).

The current work is an extension of the work of those two papers. They aim to find if accumulation of cdsRNA is a feature of TDP-43 proteinopathologies that happens beyond C9ORF72 cases. They investigated if cdsRNA occur in brains of AD patients that have TDP-43 pathology and show convincing data that it does. Next, they incorporated TDP-43 staging into their previously used ML model and found that genes that are associated with Jak1/2 inhibitors and had provided a signal for all AD in their previous work were actually specifically detecting just the TDP-43 positive cases.

Previous work has shown that knockdown of TDP-43 leads to accumulation of dsRNA (Saldi et al. EMBOJ 2014) in human cells. The presence of TDP-43 pathology in AD and the evidence of loss of function of TDP-43 has been previously shown. Therefore, it is somewhat expected that dsRNA would be present in AD brains with TDP-43 pathology. What was unknown and could not be predicted is if the presence of dsRNA in AD would elicit a significant Interferon response that could be detected between AD cases with and without TDP-43 pathology. The paper uses their ML model to suggest that this is the case, but more evidence could be shown of their existing data that could make this clearer for the reader (point 4).

Previous work from the authors suggested that Jak inhibitors could protect from dsRNA toxicity. Using CRISPR screens they were able to show that protection from toxicity is mainly elicited by the Jak family member TYK2. This is an important discovery because direct inhibition of TYK2 would probably be more tolerable to patients than inhibitors that broadly inhibit Jak family members.

Finally, they provide some evidence that dsRNA induce CXCL10 signalling which could potentially be used as a biomarker for IFN-I signalling in AD.

Major points

1. The authors demonstrate that in control cases without AD or TDP-43 pathology there is no or low presence of cdsRNA. In the brains of patients with AD and TDP-43 pathology there is accumulation of cdsRNA. However, what is lacking is a significant number of AD patients without TDP-43 pathology to make a comparison on cdsRNA levels in AD with and without TDP-43. This is an important point for the authors to make because they go on to segregate AD patients by TDP-43 pathology and suggests that this is a consequence of ISG gene expression induced by cdsRNA.

2. Line 188. The posterior cingulate cortex was the region chosen from the ROSMAP data to look for CE expression. They stated that this region is impaired early in AD progression and therefore is important for the study. However, accumulation of TDP-43 and onset of AD probably represent two distinct pathways that are co-occurring in disease and both contribute to dementia but are not mechanistic dependent on each other. Previous work has shown that in AD patients the amygdala is the first region in the brain to have TDP-43 followed by the hippocampus. Using the sequencing data from these regions

would provide for more reliable detection of CE. The ROSMAP data is obtained from Rush University where there is a framework for staging TDP-43 pathology in patients. (https://www.radc.rush.edu/docs/var/detail.htm?category=Pathology&subcategory=TDP-43&variable=tdp_st4). Why wasn't the Rush staging scaled used? Also, brain samples were obtained from ROSMAP at RUSH. They would have TDP-43 pathology on all the samples that were sequence or a large fraction of them. The analysis of CE in RNA-Seq data can be useful for new datasets, but why ignore the high-quality pathology data that most likely exist at Rush? The authors should include this staging data based on pathology at Rush to validate their CE pipeline.

3. MG132 treatment of cells is not a model of TDP-43 pathology. MG132 does lead to aggregation of TDP-43 among countless other impacts on the cell. The increase in CXCL10 cannot be attributed to TDP-43 aggregation. Most of the cells in this treatment are probably dead at 48 hours. In extended data figure 11 there is a lack of DAPI staining after MG132 treatment probably because the cells have undergone apoptosis and the DNA is fragmented. Knockdown of TDP-43 has previously been shown to lead to the accumulation of dsRNA. Does knockdown of TDP-43 lead to the induction of CXCL10?

4. In my understanding of how the ML works, only a list of genes is used as input to the model and there is no input of the expected direction of the change between groups. Given that the list of genes obtained from Jak inhibition is able to distinguish TDP-43 positive and negative patients, one would expect that genes that are downregulated by Jak inhibition would be upregulated in the population that has cdsRNA activation of Jak signalling. This cannot be inferred from the predicative potential of the gene list in the ML model. It only lets you know that there is a difference in gene expression between two groups but not that it goes in the expected direction. Could the authors provide a heatmap or other graph that shows that Jak dependent genes are upregulated in the TDP-43 positive group and not in the TDP-43 negative group. This would provide some additional evidence that the cdsRNA are not only present in the TDP-43 positive cases but induce the expected interferon signalling.

5. Figure 1c. Cell models of TDP-43 only show weak TDP-43 loss of function and barely detectable levels of aggregate TDP-43. Is the level of dsRNA accumulation in TDP-43 G298S cell line enough to activate PRR signalling?

6. For statistically analysis, did the authors perform tests to ensure that datapoints are normally distributed before conducting t-tests and ANOVA?

7. Figure 4 would benefit from more data. It is probably too much to ask for MSD of CXCL10 on AD patients biofluids. However, is there any indication in existing patients datasets (RNA, MassSpec) that CXCL10 is upregulated in AD?

Minor points.

1. Include the clinical information for all patients used for histology analysis in Extended Data Table 1
2. Figure 1C legend states that 2/3 differentiated batches were analysed but there are three different shapes in the graph. Should it just state that 3 batches were used.
3. Use of two forms of the UNC13A cryptic exon is not a good measure for TDP-43 severity. The two forms occur together and the detection of one vs both in sequencing data is probably more related to sequencing depth and RIN quality of RNA rather than degree of TDP-43 loss of function in the sample.
4. IFN-A activates TYK2 but doesn't cause cell death. Why does IFN-I activation of TYK2 following cdsRNA treatment lead to cell death?
5. Figure 1 legend line 636 extended data fig 3 should be extended data fig 4
6. Line 165 cryptic exon in stmn2 leads to premature polyadenylation not a premature stop codon.
7. Line 504 capitalize ALS in the citation
8. Figure 2b UNC13A diagram is not correct. There should be two splice junctions between exon 20 and the cryptic exon and one junction between the cryptic exon and exon 21.
9. line 213, 215, and 217 References to Extended Data Figure 8 and Extended Data figure 9 are not correct. The figures need to be switched.
10. line 223. Extended data figure 10 does not show that the downstream transcription of ISGs is inhibited.
11. Figure 3C There is an extra 2 in $\log_2(\text{FC})$
12. Line 260 Check the IC50 values for Jak1. The 1 nM figure is for binding of deucravacitinib pseudo kinase domain of Jak1 and not the kinase domain. Deucravacitinib binding at the pseudo kinase domain does not inhibit Jak1 activity. Using the binding value of 1 nM rather than a value obtained with a cell based model of Jak1 activation makes the drug appear less selective than it is.
13. Probably should have used a lower concentration of deucravacitinib for MS-TMT experiments.
14. Wording in introduction suggests cdsRNA is a common feature in ALS but it has only been shown for C9ORF72 cases. Please clarify text or provide citation that it happens in sporadic forms of the disease.
15. Line 747 Supplemental Table 2 should be Supplemental Table 1
16. Line 752 Brief outline of NGN2 differentiation would be useful.
17. How much lipofectamine 2000 was used for transfections in NGN2 cortical neurons, SY5Y cells, and ReN VM cells?
18. Line 807 is MGM2 supposed to be NGN2?

Reviewer #4

(Remarks to the Author)

I suggest the authors reconsider the title, as the current phrasing—particularly the segment “...in dementia with Alzheimer’s disease”—is unclear and awkwardly constructed. While the latest clinical criteria for Alzheimer’s disease (AD) acknowledge that a clinical diagnosis does not necessarily require the presence of dementia, incorporating this nuance into the abstract may not add value to this specific study.

The following sentence requires rephrasing for clarity:

"One of the pathologic features that stratify dementia patients is Alzheimer’s disease (AD), a disease defined by amyloid beta (Ab) plaques and neurofibrillary tangles (NFTs) of hyperphosphorylated tau protein, which can be associated to other proteinopathies, including cytoplasmic inclusions of phosphorylated TAR DNA-binding protein 43 (pTDP-43)."
The sentence is overly complex and could benefit from restructuring to improve readability and precision.

I do not believe the claim that TDP-43 pathology is the most common co-pathology in AD is entirely accurate. The frequency of co-pathologies depends on the population studied, and Lewy pathology is also highly prevalent among AD patients. This point should be clarified.

The finding of CXCL10 alterations as a potential biomarker for dsRNA pathology is intriguing. However, the study would be significantly strengthened if the authors could demonstrate changes in CXCL10 levels in the cerebrospinal fluid (CSF) of AD patients.

While this is an important study with potential translational value, it is crucial that the authors explicitly address that their findings are relevant primarily for patients with concomitant TDP-43 pathology. Any therapeutic interventions discussed should be contextualized as targeting effects specifically caused by TDP-43 pathology.

In the abstract, the authors should clarify whether JAK inhibitors influence neuroinflammation caused by increased dsRNA expression/accumulation of dsRNA or other factors that elevate IFN-I signalling.

The information about CERAD scoring in Extended Data Table 1 is outdated by over 20 years. The authors should use the updated 2012 NIA-AA ABC scoring system to determine the level of AD neuropathological change.

Extended Data Figure 1: I recommend specifying that the analysis was conducted on amygdala sections.

Extended Data Figure 7: I recommend clarifying the reference to the posterior cingulate cortex. Braak & Braak staging cannot be determined based solely on a single region, and it is unclear how this region is reflected in the graph.

Version 1:

Reviewer comments:

Reviewer #1

(Remarks to the Author)

The authors have addressed the main concerns sufficiently. The paper now better supports their conclusions. I have no additional concerns that would preclude publication.

Reviewer #2

(Remarks to the Author)

König et al. aim to understand the role of TDP-43 inclusions in Alzheimer’s Disease, how those inclusions may drive toxicity, and present candidate drugs for reversing that toxicity. Another goal is to develop novel biomarkers for AD using this information.

König et al. present compelling data that link TDP-43 pathology, cytosolic double-stranded RNA, and interferon induction in human Alzheimer’s Disease tissue, iPSC-derived neurons, and a neuron-like cancer cell line. They then train their drug repurposing pipeline using cryptic exon expression to identify candidate drugs for AD with TDP-43 pathology. This pipeline identified a number of tyrosine kinase inhibitors. They show that inhibition of tyrosine kinase signaling can rescue dsRNA-induced cell death by decreasing interferon signaling. Finally, they establish a link between an established TYK2 loss of function variant and decreased CXCL10 levels in human plasma.

The manuscript as revised is much improved and presents interesting data and hypotheses about TDP-43 pathology and its downstream pathological mechanisms and detection in human Alzheimer’s Disease. I only have a few minor points:

Line 145-148: The data in figure two show that there are 190 significant IFN-I related DEGs between AD and non-AD brains. I cannot find anything up to this point in the text or figures related to this dataset that support the idea that a specific molecular mechanism is causing (rather than correlated with) IFN-I induction. I would remove this sentence, as the very similar sentence in lines 212-215 is supported by the previous data.

Related to figure 2: It would be more convincing to have GSEA or other unbiased enrichment analysis performed on these data to get a sense of how large of a transcriptional signature ISGs are versus other changes in this AD dataset.

Also related to figure 2: The text is quite small on the heat map. Are there ways make this figure clearer and provide some

more biological meaning to this figure panel?

Line 169 contains a typo "to a premature polyadenylation"

Code and methods for calculating LFC and p-values for the CRISPR screen should be referenced and/or written up in the methods sections.

Fig5b-h: pSTAT1 label should be at the molecular weight of the band.

The legends of Extended Fig 1c and 2b state that these are graphs of the proportion of cells, but they are graphed as the number of total cells, please update graphs or figure legends (I believe graphing as proportion/percent of total cells analyzed would be the most straightforward).

Reviewer #3

(Remarks to the Author)

The authors have made improvements to their manuscript. Overall, I think that their claims that there is high cdsRNA in AD patients with TDP-43 pathology, TYK2 inhibition is protective against cell death induced by cdsRNA, and CXCL10 could be used as a biomarker to detect cdsRNA induced inflammation are well support and novel findings.

Response to major points revisions.

1. I would have like to see more cases, but the cases that they do include supports their claim that patients with high levels of pTDP-43 also have high levels of cdsRNAs.
2. Looking at cryptic exons in the amygdala or hippocampus may have increased the number of TDP-43 positive cases. However, the authors state they were trying to be conservative in assigning TDP-43 positive cases and they were still able to find a significant signal in their DRIAD-SP analysis. Therefore, I think their analysis is sufficient.
3. I still think MG132 is not great model of TDP-43 aggregation, but as there are no reliable alternatives it is sufficient. I have not detected CXCL10 expression in TDP-43 knockdown RNAseq datasets from NGN2-cortical neurons or SH-SY5Y cells. If TDP-43 pathology is inducing CXCL10 expression it might be from a gain of function of aggregates/mislocalization rather than loss of nuclear function. The updated immunofluorescence of MG132 treated cells looks very nice.
4. The new graph (Extended Fig5 C) answers my question.
5. NGN2 neurons with TDP-43 mutations have mild phenotypes, but the authors are still able to detect a modest increase in cdsRNA. On the question of whether cdsRNA in this model can drive PPR activation, the authors point to ref 33 to state that this cell line has increased expression of inflammatory response genes that is not completely rescued by STING inhibitors. However, looking at ref 33 figure 7a there appears to be a full rescue by STING inhibitors. The data as presented suggest cytoplasmic TDP-43 could contribute to formation of cdsRNA in AD and the authors make several points that the origin of cdsRNA in AD is unknown. As long as there is no claim that cytoplasmic TDP-43 is the sole driver of cdsRNA, then I think further experiments in this line are not needed.
6. Thank you for clarifying tests of normality.
7. Thank you for including references to data that shows increased CXCL10 levels in AD cases.

All previous minor points were adequately addresses.

New minor points.

Figure 3C. There is a stray datapoint in the STMN2 violin plot potentially from figure re-arrangements without grouping points.

Extended Data Fig 2a. Should labels for TDP-43 (in figure and legend) be pTDP-43?

Extended Data Fig 4a. Text for "STMN2 short" in the figure is overlapping.

Reviewer #4

(Remarks to the Author)

I am satisfied with the responses to the reviewer's comments.

Reviewer #5

(Remarks to the Author)

The main computational feature in this work seems to be the use of a previously published model called DRIAD and its minor extension to DRIAD-SP in selecting drugs for repurposing. First, the paper omits the explanation of DRIAD which needs to be entirely explained and justified in the methods section:

--The full method should be transparently explained (in words and equations) since it forms an important part of the computation.

--This method needs to be compared against others with different assumptions, potentially on synthetic data with ground truth.

DRIAD seems to be a somewhat convoluted method that works as follows:

- 1) Uses gene expression measurements of cell lines perturbed by drugs to get a differentially expressed list of genes (DEG).
- 2) Uses these DEG genes to train a logistic regression model (I had to go to the methods section of the previous paper to

find this) to predict Braak stages just from these gene expression levels.

3) Uses random lists of genes to predict the Braak stage to create a null distribution of predictiveness from which to compute significance.

The main improvement seems to be multi-class or ordinal rather than binary regression, which may seem minor but is still worth validating if treatments are Braak-specific.

I have several problems with this method. First the gene expression measurements on cell lines may not be reflective of measurements on actual tissue nor drug effects. Second, predicting Braak stage from these gene lists does not imply that the drug modulates the gene in the right direction. Third, it is not clear to me that random gene lists construct the correct null, indeed greater predictiveness could result from any genes in neurodegeneration-related pathways. Thus I would want this method rejustified and compared in this paper before I believe it can be of actual benefit. How does it compare to, for instance knowledge graph methods based on known genes? Fourth, the authors seem to use a ridge rather than lasso regularization, and this limits mechanistic interpretability and feature selection ability.

Reviewer #6

(Remarks to the Author)

Version 2:

Reviewer comments:

Reviewer #2

(Remarks to the Author)

All of my concerns in the previous rounds of reviews have been addressed. I believe that the data in this paper support the conclusions, and adds interesting and significant contributions to the field. Thank you to the authors for their thoughtful consideration of my critiques.

Reviewer #5

(Remarks to the Author)

I was asked to review the computational component of the paper and I find the addressing of their comments still somewhat lacking although it does not seem to be a large part of their paper.

---I asked them to write their method in its entirety, instead they left the short description in their methods section. I would appreciate an algorithm box or something of this sort.

---The method still seems ad hoc to me, maybe they can give more information on all drug perturbation data they used in DRIAD and which other drugs gave significant results. Information on specificity and precision vs recall should be given.

---Since their paper depends on experimental results more than computational results, maybe there is an orthogonal way of deriving baricitinib and ruxolitinib for testing.

I appreciate the authors compared to knowledge graph methods.

Reviewer #6

(Remarks to the Author)

POINT BY POINT RESPONSES:

REVIEWER COMMENTS

Reviewer #1 (Remarks to the Author):

Summary:

The manuscript from König et al, presents an interesting hypothesis which is partly supported by the evidence from multiple models/databases, including their own. The hypothesis suggests that cytoplasmic dsRNA is a proxy for TDP-43 pathology in AD which triggers INFalpha/beta signaling and downstream JAK/STAT pathway, which upon inhibition ameliorate neuroinflammation and cell death in AD cell models. The proposed mechanism is compelling and findings could be provide significant for the field of neurodegeneration. However, the data presented here are mostly correlative rather than causative to infer mechanism. While these data are intriguing, there are several questions that arise.

Major points:

- In fig 1b, authors' claim of not finding neurofibrillary tangles (NFTs) and Abeta plaques in AD brain samples needs to be verified by using specific antibodies/stains that identify aggregated forms of tau and Abeta, such as MC-1 antibody (Tau (MC-1) | ALZFORUM) for NFTs, or stains such as methoxy-X04 (Klunk et al; 2002 PMID: 12230326) for Abeta plaques. Also, the brain areas that were examined with immunostaining are not mentioned.

We want to clarify that we did find NFTs and Abeta plaques in the amygdala of these patients with Alzheimer's disease pathology (Fig. 1b), but that dsRNA was not spatially coincident with plaques and tangles. We used these antibodies: tau (cat. # NBP2-25162, Novus Biologicals, CO, USA); Aβ (cat. # CST-D54D2, Cell Signaling Technology) to label tangles and plaques, respectively. We clarify that the tissue is amygdala, both in the main text and figure legend (line 99, 117, 686, 688, 819, 1283, 1287, 1294)

- In Fig 1c, the intensity of all channels in the representative control panel is much lower than the TDP-43 mutant panel. In the graph, what is the dsRNA intensity normalized to?

We updated the images in Figure 1d to be more comparable in intensity. The dsRNA intensity is normalized to the dsRNA intensity in the isoTDP43+/+ control line.

- It appears that different brain regions are examined for upregulated ISGs and for CEs. Using the same brain regions would clarify the inference. For example, lines 137-140 state “The parahippocampal gyrus (PHG), a brain region that shows atrophy early on in disease progression^{39,40}, had the greatest number of upregulated ISGs, followed by the dorsolateral prefrontal cortex (DPFC) and the inferior frontal gyrus (IFG)”. However, while examining CEs in the same RNA seq database, in lines 187-189, “Consequently, we restricted our analysis to ROSMAP sequencing data from the posterior cingulate cortex (PCC), a brain region impaired early on in disease progression and therefore highly relevant for our study⁴⁵,” a different brain region is used. To make useful correlations, it will be helpful to examine these mRNA changes in the same brain areas.

Thank you for this point. We used available data to analyze ISG's and cryptic exons. We observe increased CE's in the PCC, and dorsolateral prefrontal cortex, but not in the head of the caudate. We chose to focus on PCC since it is involved early in AD. Conversely, we see increased ISG's in PCC (Extended Data Figure 5c).

- How do CEs relate to cdsRNA? Are the authors correlating them in any way? How does the statement in lines 204-206, “These findings support our hypothesis that cdsRNA associated with TDP-43 dysfunction contributes to IFN-I-mediated neuroinflammatory processes occurring in AD patients' brains”, relates to the findings regarding CEs in the same result subsection. This point should be clarified and discussed further.

We are using CE expression as a proxy of TDP-43 pathology, particularly nuclear hypofunction. In this study CE expression has no direct correlation with cdsRNA accumulation, but CE expression (PMID: 35197628, 35197626, 30643292, 30643298, 32790644, 28332094, 37605276, 38133681, 38175301, and the presence of cdsRNA (as seen in this study) are both linked to TDP-43 pathology. We added Extended Data Fig. 5c to show that interferon signaling is significantly upregulated in patients that we stratified TDP-43 positive which is likely linked to the spatially coincident cdsRNA.

- Choice of using Ren VM cell line over iPSC-derived NGN2 neurons for the CRISPR screen should be defended/explained further. It is confusing to understand why the authors would use a “neural” line that displays markers of neurons and glial cells and yet be considered “differentiated”. In contrast, using iPSC-derived NGN2 neurons will help decipher a more cell-autonomous signaling mechanism. It will strengthen the paper to validate the main findings in iNeurons as well.

We performed this genome-wide Cas9 screen using ReN VMs, which are human neuroprogenitor cells which grow well and are amenable for screens. The screen was performed after differentiation. As we published previously, these cells differentiate into mixed cultures of neurons, astrocytes and oligodendrocytes to conduct the screen. We have used 4 different cell lines (ReN VM, ReN CX, SH-SY5Y, and iPSC-derived NGN2 neurons) to validate our results from the genome-wide screen. In particular, the replication of these results in NGN2 cortical neurons helps to describe the cell autonomous mechanism. This aligns with our previous work in the Nd1 and Nd2 mouse models, where the dsRNA is expressed exclusively in neurons.

- In fig 3c and 3d, there is more than 50% cell death in polyIC treated cells. With that in mind, in fig 3e, is equal protein loaded for the Western blot? It would help to normalize pSTAT1 levels over total protein and with total STAT1 than with only actin. It would also help to see the entire blot to make a firm assessment of the results presented. The same applies to extended fig 10. These are some examples of the pSTAT1/STAT/actin WB (Boriero et al, 2020 The anti-STAT1 polyphenol myricetin inhibits M1 microglia activation and counteracts neuronal death - Boriero - 2021 - The FEBS Journal - Wiley Online Library; Zhao et al, 2022 STAT1 Contributes to Microglial/Macrophage Inflammation and Neurological Dysfunction in a Mouse Model of Traumatic Brain Injury | Journal of Neuroscience ([jneurosci.org](https://www.jneurosci.org)))

Thank you for asking for clarification on this point. The Western Blot samples are collected at 24h post treatment because we do not see cell death at this point. Hence we are loading the same protein amounts. This point is stated (line 924). Cell death occurs at 48-72h post treatment at which point we conduct the cell viability assay. We will enclose all uncropped western blots including the ones not shown but quantified.

- In extended fig 11, it will help to see higher magnification of the images to appreciate the translocation along with quantification. Use of standard total TDP-43 antibody (ProteinTech, Cat# 10782 (AA 203–209 and AA near N-terminus/R <https://rdcu.be/dYpSn>) will also help verify this result.

Thank you for this recommendation. We used the recommended antibody and updated Extended Data Fig. 9 and included quantifications. For this study we did not have the means to use a higher magnification.

- Dose-dependent curves for JAK or Tyk2 inhibitors to show gradual changes in pSTAT1 and CXCL10 levels will be more convincing.

Thank you for this suggestion. Please see Fig. 6b and Extended Data Fig. 8e+f to see this data, which we agree, is more convincing.

- Adding a pathway schematic describing the working mechanism proposed by the authors (similar to extended fig 2a) in the main figures will help assimilate the findings of the paper derived from different cell lines/ databases. Careful assessment of correlation and causation needs to be made.

Thank you for this comment. We opted to keep the primary data in the primary figures and leave the schematic in the extended figure given the limited space for each Figure. We updated the figure legend to stress that this model is a hypothesis based on previous work in other cell types. Also, we refer the reviewer to Fig. 6d as a schematic that describes the proposed mechanism supported by our studies.

- Overall, the images in the paper can be improved by using higher resolution/magnification (and quantification wherever applicable) as well as use of complementary antibodies such as total TDP-43 (ProteinTech, Cat# 10782 (AA 203–209 and AA near N-terminus/R <https://rdcu.be/dYpSn>), 6E10 antibody (monomeric Abeta and plaques, Biolegend, Cat# 803014) and Tau (HT7) antibody (Thermo Fisher Cat#MN1000).

Please see the comments above where we have addressed these suggestions.

- It will be useful for the authors to speculate on cell-autonomous and non-cell-autonomous features of their proposed mechanism in the disease with regard to IFN- λ signaling and neuroinflammation. The proposed mechanism in this manuscript could answer important questions regarding neuronal-glia crosstalk in neurodegeneration and is worth exploring further.

Thank you for this suggestion. We agree with this reviewer that the proposed mechanism supported in this study affords opportunities for both cell-autonomous (neurons) and non-cell autonomous (astrocytes, microglia, and other neurons that are adjacent or synaptic partners in neural circuits, see Fig. 5 in Ref. 22 in our Nd1 mouse model) activation of neuroinflammation. We also address this in our introduction in line 69-72. In addition, we are currently

writing a review article developing these ideas by incorporating work from other studies that we can't include in this paper due to space / reference citation restrictions.

Minor points:

- Line 197, “tau” should be replaced with “TDP-43”

This is the sentence in the manuscript: “To determine whether predicted TDP-43 positivity was associated with later Braak stages, we used an ordinal regression model and found no significant difference in Braak staging ($P = 0.27$) between “positive” and “negative” cases (Extended Data Fig. 5a), indicating that CE production was orthogonal to the stage of tau progression as quantified by Braak staging.” We intended to compare the stages of TDP-43 pathology with stages of tau pathology.

Reviewer #2 (Remarks to the Author):

König et al. aim to understand the role of TDP-43 inclusions in Alzheimer's Disease, how those inclusions may drive toxicity, and present candidate drugs for reversing that toxicity. Another goal is to develop novel biomarkers for AD using this information.

König et al. present compelling data that link TDP-43 pathology, cytosolic double-stranded RNA, and interferon induction in human patient tissue, iPSC-derived neurons, and a neuron-like cancer cell line. They then train their drug repurposing pipeline using cryptic exon expression to identify candidate drugs for AD with TDP-43 pathology. This pipeline identified a number of tyrosine kinase inhibitors. They show that inhibition of tyrosine kinase signaling can rescue dsRNA-induced cell death by decreasing interferon signaling. Finally, they establish a link between an established TYK2 loss of function variant and decreased CXCL10 levels in human plasma.

There are two major weaknesses in the paper. First, due to reasonable technical limitations, the authors use indirect measures of cdsRNA accumulation and/or TDP-43 pathology and draw conclusions that may be consistent with their hypotheses, but do not rule out others in a satisfactory way. This leads to contradictions in their proposed model between the first and last parts of the text as written. Second, they draw a number of conclusions based on their drug treatments at concentrations that are much higher (in some places >100-fold) than the IC50s reported in the paper. Thus, the specificity of their proposed mechanisms are not fully supported by their data. These two weaknesses can be addressed as follows:

Major Points:

1. The authors stratify patients and controls by pTDP-43 pathology but do not ever tell the reader what the stratification strategy is. This is especially important considering the small number of patients in this study.

Thank you for this point. At the time of this study the neuropathology service only used an antibody against TDP-43 in their clinical workflow. Our stratification strategy is outlined in Extended Data Figure 1.

2. It is important to show that TDP43 aggregation and/or cytosolic TDP43 and/or phospho-TDP43 is co-incident with dsRNA accumulation in Figures 1A and 1B. Quantitation of this would provide very strong evidence for their proposed model. How many TDP-43 positive cells are dsRNA negative and vice-versa? Does TDP43 aggregation/mislocalization induce dsRNA accumulation in cells adjacent to the cells with aggregation/mislocalization?

Thank you for this suggestion. We have provided quantification for costaining of TDP-43 and dsRNA in Extended Data Fig. 1c. The staining in adjacent cells is part of a larger project to add cell-type specific markers along with the neuropathologic antibodies. Without knowing cell-type specificity, it is not clear if adjacent cells are due to propagation or engulfment of the dsRNA/TDP-43 from dying neurons. Ultimately, those questions will be addressed in inducible mouse model systems or inducible brain organoid systems.

3. The authors propose that cdsRNA mediates toxicity at least in part via induction of interferon signaling. However, interferon induction can be due to many factors (as mentioned in lines 142-144 and 395-404 in the text), especially in AD. The connection between TDP43 pathology/cdsRNA accumulation and interferon induction would be much better supported if the data in Extended Data Figure 2 contained cdsRNA staining and a quantification of co-localization and/or intensity of pPKR, cdsRNA, and TDP43 pathology.

See updated version of Extended Data Fig. 2 where we show staining of phospho-PKR (activated PKR by dsRNA) is enriched in cells with cdsRNA and TDP-43 mislocalization. While PKR is an interferon stimulated gene product, the phosphorylated form is a marker of engagement with dsRNA.

Similarly, a comparison of transcriptomics (Figure 1D and Extended Data 4) from AD patients with and without TDP-43 pathology would make a much stronger argument in support of their model, rather than comparing AD+TDP43 pathology versus healthy controls.

We have not received the Rosmap TDP-43 pathology stratification in spite of requesting the DUA 6 months ago. However, we used CE expression as a proxy of TDP-43 pathology and read out average ISG expression in AD cases with presumed TDP-43 pathology cases vs AD cases with presumed TDP-43 negative cases and saw a significant increase in interferon signaling (Extended Data Fig. 5c). The comparison group is not healthy controls but AD cases without presumed TDP-43 pathology

4. To prove the specificity of their hits, the authors should validate their initial CRISPR-screen phenotype with knockdown or knockout of at least IFNAR2, IRF9, and TYK2. Using these lines, they should then show that treatment with the identified tyrosine kinase inhibitors does not rescue toxicity when the above genes are knocked out or knocked down. Similar experiments should be done with knockdown of JAK2 and JAK1 to show specificity for TYK2.

Thank you for this comment. We performed a TYK2 KD as we selected this hit as the most favorable target and the focus of this paper. See Fig. 4f-h and Fig. 5g+h. Both JAK1 and JAK2 were not hits in the screen and the selectivity of deucravacitinib at 10 nM for TYK2 over JAK1 and JAK2 (see next comment) demonstrate that JAK1 and JAK2 are not contributing to cell death in this model system.

Relatedly, the authors begin treatment with deucravacitinib at 10nM (50-fold higher than the reported TYK2 IC₅₀, but also 10-fold higher than the JAK1 IC₅₀) and see almost complete rescue of the polyIC survival phenotype. Showing a concentration dependence for rescue would provide much stronger evidence for the authors' model.

See updated Fig. 5a (Deucravacitinib calculated EC₅₀ = 1.481 nM) and we also updated the manuscript with the correct IC₅₀ values for JAK 1-3 of deucravacitinib (IC₅₀ = 0.2 nM compared to IC₅₀(JAK1-3) > 10 μM).

5. The model in Figure 4D does not take into account other possible sources of neuroinflammation or the role of TDP-43 pathology, as noted in the text (lines 395-

404). The model should be updated so that it does not appear the authors are proposing that only cdsRNA can cause neuroinflammation to cause disease.

Thank you for this comment, as we don't believe that cdsRNA is the only contribution of interferon signalling in the CNS or the only cause of neuroinflammation in the brain. See updated figure legend - "Schematic overview of hypothesized pathomechanisms underlying cdsRNA in neurodegenerative diseases that focuses on dsRNA-mediated neuroinflammation. Other potential triggers for neuroinflammation are not depicted here for simplicity's sake."

6. While the data is sound, it is unclear how relevant ReN VM conditioned media are to human plasma levels of CXCL10. Providing some references that show that conditioned media can be extrapolated to human plasma biomarker levels would make the argument for CXCL10 as an AD biomarker much stronger. To directly compare the mechanisms with Figure 5a, the authors could use gene editing to make the identified SNP and measure CXCL10 levels. Alternatively, directly examining CXCL10 plasma levels for AD versus healthy controls would be sufficient.

Thank you for this comment to strengthen the connection between our in vitro observations and measurements in patient samples. Numerous studies show elevated levels of CXCL10 in AD relative to healthy controls (PMID: 29893515, 36967827, 28413983, 37291639, 29223680). This review (PMID: 37863021) states evidence that deucravacitinib mimics the functional effect of the SNP P1104A which is the genetic variant we present in Fig. 6a. We agree with this reviewer that with this evidence of plasma levels in AD patients vs. healthy controls, that creating a human REN cell line with this SNP will not substantial additional evidence to prove our point.

Minor Points:

1. For Figure 1C, it would be important to show that TDP43 is mislocalized with TDP43 IF.

We do not observe obvious cytoplasmic TDP-43 mislocalization in these mutant cells without a stressor, as is the case for other labs. We do observe increased dsRNA in the mutant line, consistent with nuclear hypofunction of TDP-43.

2. The heatmap in Extended Data 4 is much more informative than in Figure 1D

We agree and moved the heatmap to main Fig. 2a

3. A sentence or two introducing the RenVM cells would clarify those cells' relevance to the reader.

Thank you, we do this in line 220 - 223 and 285 - 288.

4. It is unclear how hits are called in the CRISPR screen. Please add this to the methods or in the results section.

Target genes were considered a hit if the LFC was equal or greater than 2.0 after rounding to the first decimal place. We added this information both to the main text and to the methods.

5. It would strengthen the authors' argument that dsRNA toxicity is mediated by interferon if they showed gene enrichment analysis (e.g. using ENRICH: <https://maayanlab.cloud/Enrichr/>) of their screen hits.

Thank you, we added this analysis to Fig. 4c.

6. It would help with clarity to have the drug targets listed with the drugs in Figure 3 and Extended Data 10.

Thank you, we added the primary drug targets with IC50 values to Extended Data Fig. 8g.

7. The authors mention that cGAS-STING activation can play a role in dsRNA-induced toxicity. If possible (not needed for publication), it would shed a huge amount of light on the relative contribution of the authors' proposed mechanism versus cGAS-STING dsRNA sensing to perform those experiments in a cell line that does express cGAS-STING.

SH-SY5Y cells express STING which we show in the new Extended Data Fig. 8h. As seen in Fig. 5e+f, there do not seem to be additional STING effects to dsRNA-mediated toxicity elicited by the dsRNA mimetic poly(I:C).

8. It may be worth discussing a recent paper that identified TYK2 (PMID 39528671) as a regulator of tau levels and turnover in the discussion section.

Thank you, we cite and discuss this paper in the discussion (line 414 - 418)

9. Generally the paper would be improved if more of the extended data were in the main figures.

Thank you for highlighting our extended data. Within the limitations of space requirements, we have moved figures from Extended Data Figures to the Main Figures.

10. I cannot offer expert feedback on the technical merits of the DRIAD-SP training or framework.

Reviewer #3 (Remarks to the Author):

Summary of the key results

Previous work by the Albers group had identified cytoplasmic double stranded RNA (cdsRNA) in the frontal and motor cortex in post-mortem tissue of ALS-FTD patients with C9ORF72 repeat expansion. CdsRNA was proposed to be produced in this model from antisense sense transcription of RNA from G4C2 expansion. CdsRNA was found to colocalize with phosphorylated TDP-43 in patient samples. (Rodriguez et al 2021 Science Translational medicine). They found cdsRNA to be toxic to neural cells and this toxicity could be rescued with treatment by FDA approved JAK1/2 inhibitors Ruxolitinib and Baricitinib.

Separately the Albers group developed a machine learning model to aid the repurposing of FDA approved drugs for use in Alzheimer's Disease (AD). The model compares gene expression changes at different AD severities with gene perturbations induced in neural cells following treatment with FDA approved drugs. They found the gene list that comprises perturbed genes follow Jak1/2 inhibition by ruxolitinib was able to reliably predict early vs late disease stages in addition to inhibitors for kinases in the ULK and NEK families. (Rodriguez et al. 2021 Nature Communications).

The current work is an extension of the work of those two papers. They aim to find if accumulation of cdsRNA is a feature of TDP-43 proteinopathologies that happens beyond C9ORF72 cases. They investigated if cdsRNA occur in brains of AD patients that have TDP-43 pathology and show convincing data that it does. Next, they incorporated TDP-43 staging into their previously used ML model and found that genes that are associated with Jak1/2 inhibitors and had provided a signal for all AD

in their previous work were actually specifically detecting just the TDP-43 positive cases.

Previous work has shown that knockdown of TDP-43 leads to accumulation of dsRNA (Saldi et al. EMBOJ 2014) in human cells. The presence of TDP-43 pathology in AD and the evidence of loss of function of TDP-43 has been previously shown. Therefore, it is somewhat expected that dsRNA would be present in AD brains with TDP-43 pathology. What was unknown and could not be predicted is if the presence of dsRNA in AD would elicit a significant Interferon response that could be detected between AD cases with and without TDP-43 pathology. The paper uses their ML model to suggest that this is the case, but more evidence could be shown of their existing data that could make this clearer for the reader (point 4).

Previous work from the authors suggested that Jak inhibitors could protect from dsRNA toxicity. Using CRISPR screens they were able to show that protection from toxicity is mainly elicited by the Jak family member TYK2. This is an important discovery because direct inhibition of TYK2 would probably be more tolerable to patients than inhibitors that broadly inhibit Jak family members.

Finally, they provide some evidence that dsRNA induce CXCL10 signalling which could potentially be used as a biomarker for IFN-I signalling in AD.

Major points

1. The authors demonstrate that in control cases without AD or TDP-43 pathology there is no or low presence of dsRNA. In the brains of patients with AD and TDP-43 pathology there is accumulation of dsRNA. However, what is lacking is a significant number of AD patients without TDP-43 pathology to make a comparison on dsRNA levels in AD with and without TDP-43. This is an important point for the authors to make because they go on to segregate AD patients by TDP-43 pathology and suggests that this is a consequence of ISG gene expression induced by dsRNA.

Thank you for this point. We do not claim that TDP-43 negative cases do not show dsRNA. However as seen in Extended Data Fig. 1a we observe dsRNA much more frequently in AD cases with severe TDP-43 cytoplasmic pathology in the amygdala in our study (line 103 - 104) which admittedly has a limited number of assessed cases of mild and moderate TDP-43 cytoplasmic pathology in the amygdala. dsRNA can arise from many genetic sources

(nucleus, mitochondria) and other mechanisms and therefore is not relying on TDP-43 cytoplasmic pathology (nuclear hypofunction of TDP-43 giving rise to cryptic exons). We added this to our discussion (line 423 - 426). In our introduction we do point out that the cdsRNA-mediated innate immune-mediated toxicity can arise independently from TDP-43 cytoplasmic pathology (line 72 - 74).

2. Line 188. The posterior cingulate cortex was the region chosen from the ROSMAP data to look for CE expression. They stated that this region is impaired early in AD progression and therefore is important for the study.

We observed an enrichment of CE's in the PCC, but not in the head of the caudate or the dorsolateral prefrontal cortex in the available ROSMAP data. We chose to focus on the Rosmap data set since that data set was generated using paired-end sequencing, which is more sensitive for CE expression as compared to brain regions from MSBB with single end sequencing – as shown in Extended Data Fig. 4a.

However, accumulation of TDP-43 and onset of AD probably represent two distinct pathways that are co-occurring in disease and both contribute to dementia but are not mechanistic dependent on each other. Previous work has shown that in AD patients the amygdala is the first region in the brain to have TDP-43 followed by the hippocampus. Using the sequencing data from these regions would provide for more reliable detection of CE.

We did not analyze data from the amygdala or the hippocampus, but we did look at HCN and DLPFC in which we saw no enrichment of CE's in AD cases.

The ROSMAP data is obtained from Rush University where there is a framework for staging TDP-43 pathology in patients.

(https://www.radc.rush.edu/docs/var/detail.htm?category=Pathology&subcategory=TDP-43&variable=tdp_st4). Why wasn't the Rush staging scaled used?

DRIAD-SP is based on Braak staging, not TDP-43 staging. CE expression is used to stratify input data and compare outcomes, but not to train the whole model. We look forward to building a DRIAD-TDP-43 once that data becomes available to us, but the correlation with disease progression between tau pathology and clinical outcomes is not as established for TDP-43 pathology.

Also, brain samples were obtained from ROSMAP at RUSH. They would have TDP-43 pathology on all the samples that were sequenced or a large fraction of them. The analysis of CE in RNA-Seq data can be useful for new datasets, but why ignore the high-quality pathology data that most likely exist at Rush? The authors should include this staging data based on pathology at Rush to validate their CE pipeline.

Thank you for this excellent suggestion. We requested the information yet still have not received after 6 months due to some issues with the execution of the DUA between the MGH and Rush for inexplicable reasons.

3. MG132 treatment of cells is not a model of TDP-43 pathology. MG132 does lead to aggregation of TDP-43 among countless other impacts on the cell. The increase in CXCL10 cannot be attributed to TDP-43 aggregation. Most of the cells in this treatment are probably dead at 48 hours. In extended data figure 11 there is a lack of DAPI staining after MG132 treatment probably because the cells have undergone apoptosis and the DNA is fragmented. Knockdown of TDP-43 has previously been shown to lead to the accumulation of dsRNA. Does knockdown of TDP-43 lead to the induction of CXCL10?

We did this experiment to determine if TDP-43 mislocalization was sufficient to drive interferon signaling (see updated Extended Data Fig. 9) and if that innate signaling was blocked by FDA-approved JAK and TYK2 inhibitors. Others have used MG-132 to induce TDP-43 mislocalization (PMID: 32040555, 21829535). We now include quantifications of TDP-43 cytoplasmic mislocalization after treatment with MG-132 with two different antibodies against TDP-43. We agree with the reviewer that MG-132 has pleiomorphic effects on these cells beyond TDP-43 mislocalization – an issue with other stressors used to evoke TDP-43 mislocalization. The cells are not dead within 48h (and DAPI is observed in the nucleus), but do start dying within 72h. It may take longer than 72h for dsRNA to accumulate to detectable levels (as seen in the TDP43 mutant iPSC lines). We were able to show that CXCL10 protein was induced by MG-132 and this cytokine induction was prevented by the JAK and TYK2 inhibitors in differentiated human ReN cells. TDP-43 knockdown leads to accumulation of dsRNA and CXCL10 production in primary rat astrocytes (ref. 21). We observe in our screen that 3 out of 4 different guide RNAs against mouse TARDBP slightly exacerbated cell death 48h after poly I:C treatment, but have not validated that result in a dedicated experiment.

4. In my understanding of how the ML works, only a list of genes is used as input to the model and there is no input of the expected direction of the change between groups. Given that the list of genes obtained from Jak inhibition is able to distinguish TDP-43 positive and negative patients, one would expect that genes that are downregulated by Jak inhibition would be upregulated in the population that has dsRNA activation of Jak signalling. This cannot be inferred from the predicative potential of the gene list in the ML model. It only lets you know that there is a difference in gene expression between two groups but not that it goes in the expected direction.

Correct, results from DRIAD-SP need to be validated by in vitro and in vivo experiments, which we did with the JAK inhibitors.

Could the authors provide a heatmap or other graph that shows that Jak dependent genes are upregulated in the TDP-43 positive group and not in the TDP-43 negative group. This would provide some additional evidence that the dsRNA are not only present in the TDP-43 positive cases but induce the expected interferon signalling.

We included a simplified version of this heatmap in Extended Data Fig. 5c in which we show that average ISG expression is significantly upregulated in cases that we stratified as TDP-43 positive based on the detection of cryptic exons.

5. Figure 1c. Cell models of TDP-43 only show weak TDP-43 loss of function and barely detectable levels of aggregate TDP-43. Is the level of dsRNA accumulation in TDP-43 G298S cell line enough to activate PRR signalling?

As noted earlier, we do not see TDP43 mislocalization in the NGN2 iPSC derived neural cells. However, the STING antagonist didn't fully inhibit IFN-signaling in NGN2 derived iPSCs (ref. 33), so we attribute the rest to dsRNA accumulation triggering RIG-I / MDA5 activation.

6. For statistical analysis, did the authors perform tests to ensure that datapoints are normally distributed before conducting t-tests and ANOVA?

Yes, a normality test (either Shapiro-Wilk, D'Agostino-Pearson omnibus, Anderson-Darling, or Kolmogorov-Smirnov test with Dallal-Wilkinson-Lilliefors P value where appropriate and applicable based on replicate number) was conducted with a significance level of 0.05 after outlier testing and exclusion (ROUT Q = 1% and Grubbs alpha = 0.05). If there was no difference in

significance with and without the respective outliers, the outliers were still included in the figure unless otherwise pointed out in the figure legend to ensure that as few values are excluded as possible for a comprehensive representation of the data. In the few cases where normality tests failed, a Wilcoxon Rank Sum test (which does not assume normal distribution) was conducted instead of a t test which resulted in the same significance as the t test.

7. Figure 4 would benefit from more data. It is probably too much to ask for MSD of CXCL10 on AD patients biofluids. However, is there any indication in existing patients datasets (RNA, MassSpec) that CXCL10 is upregulated in AD?

Thank you for this comment. See reviewer 2 major point #6. Various studies show elevated levels of CXCL10 in AD (PMID: 29893515, 36967827, 28413983, 37291639, 29223680).

Minor points.

1. Include the clinical information for all patients used for histology analysis in Extended Data Table 1

See updated Extended Data Table 1

2. Figure 1C legend states that 2/3 differentiated batches were analysed but there are three different shapes in the graph. Should it just state that 3 batches were used.

Correct, thank you. Figure legend was adjusted.

3. Use of two forms of the UNC13A cryptic exon is not a good measure for TDP-43 severity. The two forms occur together and the detection of one vs both in sequencing data is probably more related to sequencing depth and RIN quality of RNA rather than degree of TDP-43 loss of function in the sample.

Both CE isoforms can occur in the same cell but differences in cell types may alter their relative distribution. We added Extended Data Fig. 4b to show that all three cryptic exons are detected independently in the majority of our data set (see small fraction of samples where CE1 and CE2 are detected together). Only 7/546 (1.3%) cases were deemed positive by having only CE1 and CE2 without STMN2 CE. We agree that differences in detection of the CE isoforms could be related to cell type specificity, sequencing depth, or RIN. However,

our data indicates that searching for both isoforms is important to capture the full CE landscape from UNC13A.

4. IFN-A activates TYK2 but doesn't cause cell death. Why does IFN-I activation of TYK2 following cdsRNA treatment lead to cell death?

This is an interesting and subtle point. While TYK2 function appears to be necessary for cell death mediated by dsRNA, IFN-I activation alone is not sufficient, suggesting that dsRNA is having additional actions beyond interferon induction to mediate cell death. Ongoing experiments are addressing the additional mechanisms evoked by dsRNA but are beyond the scope of this paper. We try to capture this point by stating "in a subsequent validation it became clear that TYK2 is necessary to elicit human neural cell death" to abstract (line 43 - 44) and "The mechanism of cell death linked to dsRNA is TYK2-dependent, and IFN-I signaling alone (although it activates TYK2) is not sufficient to trigger cell death, indicating that dsRNA has broader effects than just triggering interferon, such as the modulation of autophagy⁷⁶ - perhaps acting through TBK1" (line 365 - 369) to the discussion.

5. Figure 1 legend line 636 extended data fig 3 should be extended data fig 4

Correct, thank you. Adjusted.

6. Line 165 cryptic exon in stmn2 leads to premature polyadenylation not a premature stop codon.

Correct, thank you. Adjusted.

7. Line 504 capitalize ALS in the citation

Thank you. Done.

8. Figure 2b UNC13A diagram is not correct. There should be two splice junctions between exon 20 and the cryptic exon and one junction between the cryptic exon and exon 21.

Correct, thank you. Adjusted.

9. line 213, 215, and 217 References to Extended Data Figure 8 and Extended Data figure 9 are not correct. The figures need to be switched.

Correct, thank you. Adjusted.

10. line 223. Extended data figure 10 does not show that the downstream transcription of ISGs is inhibited.

Correct. We cited previous work showing that downstream transcription of ISGs is inhibited.

11. Figure 3C There is an extra 2 in log₂(FC)

Thank you, corrected.

12. Line 260 Check the IC₅₀ values for Jak1. The 1 nM figure is for binding of deucravacitinib pseudo kinase domain of Jak1 and not the kinase domain. Deucravacitinib binding at the pseudo kinase domain does not inhibit Jak1 activity. Using the binding value of 1 nM rather than a value obtained with a cell based model of Jak1 activation makes the drug appear less selective than it is.

Thank you for pointing this out, this was indeed a mistake. Values have been corrected: IC₅₀ = 0.2 nM compared to IC₅₀(JAK1-3) > 10 μM

13. Probably should have used a lower concentration of deucravacitinib for MS-TMT experiments.

In retrospect, we agree that a lower dose would have likely been informative with less off target effects. At the time, a previous screen in undifferentiated cells had suggested dose dependency at that concentration of deucravacitinib which is why this concentration was selected.

14. Wording in introduction suggests cdsRNA is a common feature in ALS but it has only been shown for C9ORF72 cases. Please clarify text or provide citation that it happens in sporadic forms of the disease.

Thank you, we added this clarification in the last sentence of the abstract. In the introduction we already mention that this applies to ALS and FTD patients with an intronic expansion of hexanucleotide repeats in the C9ORF72 gene (line 69 - 72).

15. Line 747 Supplemental Table 2 should be Supplemental Table 1

Correct, thank you. Adjusted.

16. Line 752 Brief outline of NGN2 differentiation would be useful.

Agree, we added the description to the methods (line 866 - 874).

17. How much lipofectamine 2000 was used for transfections in NGN2 cortical neurons, SY5Y cells, and ReN VM cells?

Dilution factor 3:1000 - added to methods.

18. Line 807 is MGM2 supposed to be NGN2?

Correct, thank you. Adjusted.

Reviewer #4 (Remarks to the Author):

I suggest the authors reconsider the title, as the current phrasing—particularly the segment “...in dementia with Alzheimer’s disease”—is unclear and awkwardly constructed. While the latest clinical criteria for Alzheimer’s disease (AD) acknowledge that a clinical diagnosis does not necessarily require the presence of dementia, incorporating this nuance into the abstract may not add value to this specific study.

We agree and have changed the title to remove the word “dementia” and focused on Alzheimer’s disease pathology.

The following sentence requires rephrasing for clarity:

"One of the pathologic features that stratify dementia patients is Alzheimer’s disease (AD), a disease defined by amyloid beta (Ab) plaques and neurofibrillary tangles (NFTs) of hyperphosphorylated tau protein, which can be associated to other proteinopathies, including cytoplasmic inclusions of phosphorylated TAR DNA-binding protein 43 (pTDP-43)."

The sentence is overly complex and could benefit from restructuring to improve readability and precision.

Thank you, the sentence has been adjusted to “Alzheimer’s disease (AD), a disease defined by amyloid beta (Ab) plaques and neurofibrillary tangles (NFTs) of hyperphosphorylated tau protein, can be associated to other proteinopathies, including cytoplasmic inclusions of phosphorylated TAR DNA-binding protein 43 (pTDP-43)”

I do not believe the claim that TDP-43 pathology is the most common co-pathology in AD is entirely accurate. The frequency of co-pathologies depends on the population studied, and Lewy pathology is also highly prevalent among AD patients. This point should be clarified.

We modified this sentence to state that TDP-43 is one of the most prominent proteinopathies in AD (line 63 - 64). We quote this source PMID: 34930382

The finding of CXCL10 alterations as a potential biomarker for dsRNA pathology is intriguing. However, the study would be significantly strengthened if the authors could demonstrate changes in CXCL10 levels in the cerebrospinal fluid (CSF) of AD patients.

See response to Reviewer 2 major point #6 and reviewer 3 major point #7. Various studies show elevated levels of CXCL10 in AD (PMID: 29893515, 36967827, 28413983, 37291639, 29223680).

While this is an important study with potential translational value, it is crucial that the authors explicitly address that their findings are relevant primarily for patients with concomitant TDP-43 pathology. Any therapeutic interventions discussed should be contextualized as targeting effects specifically caused by TDP-43 pathology.

Thank you for this comment, and we agree that these insights will be relevant to address the heterogeneity of patients with Alzheimer’s disease pathology to work towards a precision medicine approach. While our data especially our cell-based models with innate immune activation of IFN-I signaling does not require TDP-43 cytoplasmic inclusions, we suggest that the translational potential of our findings are relevant to patients with concomitant TDP-43 cytoplasmic inclusion pathology (which is currently not accessible in living patients). In addition, we suggest that TYK2 inhibition should be studied in any neurodegenerative disease with accumulation of immunogenic cytoplasmic dsRNA, possibly due to lack of TDP-43 function in the nucleus, which often correlates with TDP43 cytoplasmic pathology. Thus, at the end of

the abstract we focus the translational potential on neuroinflammatory mechanisms that are present in TDP43-associated Alzheimer's disease pathology and at the end of the discussion we are focused on TDP43 pathology, which is both ectopic cytoplasmic TDP43 and nuclear hypofunction of TDP43 as detected by cryptic exons.

In the abstract, the authors should clarify whether JAK inhibitors influence neuroinflammation caused by increased dsRNA expression/accumulation of dsRNA or other factors that elevate IFN-I signalling.

Our data cannot rule out that JAK inhibitors could block innate immune signaling triggered by type I interferon. The damage associated molecular patterns (DAMPs) most closely associated with IFN-I are dsRNA (the focus of this paper) and dsDNA (cGas/STING), which we mention in the discussion (lines 375 - 384). In addition, the source of dsRNA and dsDNA include the nuclear genome, mitochondrial genome, and viruses.

The information about CERAD scoring in Extended Data Table 1 is outdated by over 20 years. The authors should use the updated 2012 NIA-AA ABC scoring system to determine the level of AD neuropathological change.

Thank you, agree, we have updated the data that was originally reported to the ABC scoring system. See updated Extended Data Table 1.

Extended Data Figure 1: I recommend specifying that the analysis was conducted on amygdala sections.

Agree, amygdala has been specified.

Extended Data Figure 7: I recommend clarifying the reference to the posterior cingulate cortex. Braak & Braak staging cannot be determined based solely on a single region, and it is unclear how this region is reflected in the graph.

This is now Extended Data Fig. 5a. The figure legend clarifies that our TDP-43 classification is based on detectable CE's in the PCC. We wanted to assess if there were any differences in Braak staging between our binary CE (TDP-43) strata, i.e., to rule out that patients with detectable CE were at later stages of the disease (based on their overall Braak stage).

REVIEWER COMMENTS

Reviewer #1 (Remarks to the Author):

The authors have addressed the main concerns sufficiently. The paper now better supports their conclusions. I have no additional concerns that would preclude publication.

Thank you very much for your time and thoughtful feedback. We're pleased to know that our response addressed your concerns and that you found it satisfactory.

Reviewer #2 (Remarks to the Author):

König et al. aim to understand the role of TDP-43 inclusions in Alzheimer's Disease, how those inclusions may drive toxicity, and present candidate drugs for reversing that toxicity. Another goal is to develop novel biomarkers for AD using this information.

König et al. present compelling data that link TDP-43 pathology, cytosolic double-stranded RNA, and interferon induction in human Alzheimer's Disease tissue, iPSC-derived neurons, and a neuron-like cancer cell line. They then train their drug repurposing pipeline using cryptic exon expression to identify candidate drugs for AD with TDP-43 pathology. This pipeline identified a number of tyrosine kinase inhibitors. They show that inhibition of tyrosine kinase signaling can rescue dsRNA-induced cell death by decreasing interferon signaling. Finally, they establish a link between an established TYK2 loss of function variant and decreased CXCL10 levels in human plasma.

The manuscript as revised is much improved and presents interesting data and hypotheses about TDP-43 pathology and its downstream pathological mechanisms and detection in human Alzheimer's Disease. I only have a few minor points:

Line 145-148: The data in figure two show that there are 190 significant IFN-I related DEGs between AD and non-AD brains. I cannot find anything up to this point in the text or figures related to this dataset that support the idea that a specific molecular mechanism is causing (rather than correlated with) IFN-I induction. I would remove this sentence, as the very similar sentence in lines 212-215 is supported by the previous data.

Thank you for this helpful suggestion. We agree that removing this sentence is appropriate, as it helps avoid unnecessary repetition.

Related to figure 2: It would be more convincing to have GSEA or other unbiased enrichment analysis performed on these data to get a sense of how large of a transcriptional signature ISGs are versus other changes in this AD dataset.
Also related to figure 2: The text is quite small on the heat map. Are there ways make this figure clearer and provide some more biological meaning to this figure panel?

We performed an unbiased enrichment analysis using our list of ISGs and compared it with random gene lists from the KEGG MEDICUS database. Our results show that the ISG list remains the most enriched in three out of five brain regions in AD patients. To add biological context to Figure 2, we included two new panels (Fig. 2a,b) illustrating this observation, which also highlights that the ISG list is generally enriched across all assessed brain regions compared with other gene lists.

Line 169 contains a typo “to a premature polyadenylation”

We corrected the mistake. Thank you!

Code and methods for calculating LFC and p-values for the CRISPR screen should be referenced and/or written up in the methods sections.

Thank you for pointing this out. We clarified it in the method section (see line 1051 - 1052) - the code and method can be found in reference 119 which uses the algorithm “STARS” that can also be found under this link provided by the BROAD Institute.

Fig5b-h: pSTAT1 label should be at the molecular weight of the band.

Fixed. Thank you!

The legends of Extended Fig 1c and 2b state that these are graphs of the proportion of cells, but they are graphed as the number of total cells, please update graphs or figure legends (I believe graphing as proportion/percent of total cells analyzed would be the most straightforward).

Thank you for this suggestion. We adjusted the figures accordingly and mentioned the total cell count in the figure legends.

Reviewer #3 (Remarks to the Author):

The authors have made improvements to their manuscript. Overall, I think that their claims that there is high cdsRNA in AD patients with TDP-43 pathology, TYK2 inhibition is protective against cell death induced by cdsRNA, and CXCL10 could be used as a biomarker to detect cdsRNA induced inflammation are well support and novel findings.

Response to major points revisions.

1. I would have like to see more cases, but the cases that they do include supports their claim that patients with high levels of pTDP-43 also have high levels of cdsRNAs.

Thank you for this comment. We agree that more cases are always desirable, but our findings are robust.

2. Looking at cryptic exons in the amygdala or hippocampus may have increased the number of TDP-43 positive cases. However, the authors state they were trying to be conservative in assigning TDP-43 positive cases and they were still able to find a significant signal in their DRIAD-SP analysis. Therefore, I think their analysis is sufficient.

Correct — we intentionally took a more conservative approach, and we appreciate your positive feedback.

3. I still think MG132 is not great model of TDP-43 aggregation, but as there are no reliable alternatives it is sufficient. I have not detected CXCL10 expression in TDP-43 knockdown RNAseq datasets from NGN2-cortical neurons or SH-SY5Y cells. If TDP-43 pathology is inducing CXCL10 expression it might be from a gain of function of aggregates/mislocalization rather than loss of nuclear function. The updated immunofluorescence of MG132 treated cells looks very nice.

We are pleased that we were able to address your request, and we greatly appreciated your antibody recommendation in the previous round of revisions.

Thank you also for sharing your observations from NGN2-cortical neurons and SH-SY5Y; these insights will be very helpful in guiding future studies.

4. The new graph (Extended Fig5 C) answers my question.

Thank you for confirming!

5. NGN2 neurons with TDP-43 mutations have mild phenotypes, but the authors are still able to detect a modest increase in cdsRNA. On the question of whether cdsRNA in this model can drive PPR activation, the authors point to ref 33 to state that this cell line has increased expression of inflammatory response genes that is not completely rescued by STING inhibitors. However, looking at ref 33 figure 7a there appears to be a full rescue by STING inhibitors. The data as presented suggest cytoplasmic TDP-43 could contribute to formation of cdsRNA in AD and the authors make several points that the origin of cdsRNA in AD is unknown. As long as there is no claim that cytoplasmic TDP-43 is the sole driver of cdsRNA, then I think further experiments in this line are not needed.

Thank you for pointing this out. Yes, we do not claim that TDP-43 is the sole driver of cdsRNA. We added a clarification in line 424 - 425 and the subsequent paragraph in our discussion also highlights other potential sources and drivers of cdsRNA.

6. Thank you for clarifying tests of normality.

7. Thank you for including references to data that shows increased CXCL10 levels in AD cases.

Happy to!

All previous minor points were adequately addresses.

Thank you for your confirmation. We are glad to hear this.

New minor points.

Figure 3C. There is a stray datapoint in the STMN2 violin plot potentially from figure re-arrangements without grouping points.

Fixed - Thank you!

Extended Data Fig 2a. Should labels for TDP-43 (in figure and legend) be pTDP-43?

The labels are indeed correct and supposed to be TDP-43, but we understand why the reviewer is asking this as we used pTDP-43 in Figure 1. In Extended Data Fig. 2, in which we analyzed immunogenicity of dsRNA, we did not apply pTDP-43 antibody at the time. However, since TDP-43 is also detecting cytoplasmic pTDP-43 inclusions and since this panel shows that the TDP-43 signal is cytoplasmic as well, we assume that this marker is sufficiently showing ectopic cytoplasmic TDP-43 and spatial coincidence with immunogenic dsRNA even without specifically detecting the phosphorylated version of TDP-43.

Extended Data Fig 4a. Text for “STMN2 short” in the figure is overlapping.

Fixed - Thank you!

Reviewer #4 (Remarks to the Author):

I am satisfied with the responses to the reviewer's comments.

Thank you very much for your time. We are glad to hear we could address your comments and appreciated your valuable input.

Reviewer #5 (Remarks to the Author):

The main computational feature in this work seems to be the use of a previously published model called DRIAD and its minor extension to DRIAD-SP in selecting drugs for repurposing. First, the paper omits the explanation of DRIAD which needs to be entirely explained and justified in the methods section:

--The full method should be transparently explained (in words and equations) since it forms an important part of the computation.

The methods and code for DRIAD-SP are explained in the method section in its entirety. However, we appreciate this comment as it became clear to us that we did not label the sections transparently enough. We now included DRIAD-SP in the title of each method section that refers to this part of the manuscript.

--This method needs to be compared against others with different assumptions, potentially on synthetic data with ground truth.

DRIAD-SP (formerly referred to as DRIAD, prior to the incorporation of ordinal ridge regression and tailoring for TDP-43–positive patients) is a validated and published machine-learning pipeline. In its original publication (<https://doi.org/10.1038/s41467-021-21330-0>), DRIAD was benchmarked against other datasets. For instance, Figure 1c illustrates how different datasets were processed through DRIAD to test and compare various assumptions.

DRIAD seems to be a somewhat convoluted method that works as follows:

- 1) Uses gene expression measurements of cell lines perturbed by drugs to get a differentially expressed list of genes (DEG).
- 2) Uses these DEG genes to train a logistic regression model (I had to go to the methods section of the previous paper to find this) to predict braak stages just from these gene expression levels.
- 3) Uses random lists of genes to predict the braak stage to create a null distribution of predictiveness from which to compute significance.

The main improvement seems to be multi-class or ordinal rather than binary regression, which may seem minor but is still worth validating if treatments are braak-specific.

I have several problems with this method. First the gene expression measurements on cell lines may not be reflective of measurements on actual tissue nor drug effects.

We agree with the reviewer that drug repurposing predictors such as DRIAD-SP do not directly predict drug effects in humans. The purpose of our approach is to nominate potential repurposing candidates based on gene expression signatures from human cells treated with the respective compounds. These candidates must then undergo validation in vitro and, if promising, in vivo before any conclusions can be drawn about their effects in tissue. We wish to emphasize that our claims are limited to this framework and do not extend beyond it.

Second, predicting Braak stage from these gene lists does not imply that the drug modulates the gene in the right direction.

Correct. As stated in DRIAD's original publication (<https://doi.org/10.1038/s41467-021-21330-0>), both the introduction and the discussion explicitly clarify that the direction of a nomination is not determined—that is, the approach does not predict whether a compound will improve or worsen the disease phenotype. Rather, the goal is to nominate candidates whose relevance must then be tested through in-vitro experiments and, if promising, in-vivo validation. To underscore this point, we have added a clarifying sentence in the current manuscript (see line 159 - 161).

Third, it is not clear to me that random gene lists construct the correct null, indeed greater predictiveness could result from any genes in neurodegeneration-related pathways. Thus I would want this method rejustified and compared in this paper before I believe it can be of actual benefit. How does it compare to, for instance knowledge graph methods based on known genes?

We thank the reviewer for this comment. Indeed our hypothesis about how DRIAD-SP works is based on the idea that drugs which perturb a large number of neurodegeneration-related pathway members will yield a screen hit when compared to random gene lists. The key advantage of this approach is its unbiased nature. By screening compounds in differentiated ReNcell VM cells and comparing drug-perturbed gene signatures directly against patient transcriptional data, we avoid relying solely on existing disease pathway annotations, which may be incomplete or biased toward well-studied mechanisms.

Nonetheless, we attempted to compare DRIAD-SP's results to established knowledge graph (KG) methods. We performed a comparison using publicly available results from two sources:

- 1. TxGNN: A state-of-the-art graph neural network model that predicts drug-disease relationships from a large biomedical knowledge graph. We used its published list of 200 ranked drug candidates for AD. [doi:10.1038/s41591-024-03233-x]*
- 2. ESCARGOT on AlzKB: A LLM based method identifying drug candidates associated with notes related to "AD Pathway" (20 drugs) and "AD BodyPart" (39 drugs) pathways in AlzKB. [doi:10.1186/s13040-025-00466-5]. Out of these 46 unique drugs, four were identified as novel repurposing candidates.*

The predictions from our data-driven DRIAD-SP model and the knowledge-based TxGNN model show very little overlap. TxGNN identifies a large number of established (approved in the early 2000's) but largely ineffective AD drugs of the cholinesterase inhibitor class (e.g., donepezil), but does not

prioritize JAK inhibitor compounds identified by DRIAD-SP. Of the top 200 drugs prioritized by TxGNN for AD, only one (fedratinib) is a JAK inhibitor at rank 197. ESCARGOT on AlzKB identified four novel compounds without previous AD association: three antineoplastic agents (epirubicin, vemurafenib, fulvestrant) and vitamin A. There was minimal overlap between the TxGNN and ESCARGOT drug sets, with only 1 of 20 "AD Pathway" drugs and 3 of 39 "AD BodyPart" drugs appearing in the TxGNN results. None of the 46 drugs prioritized by ESCARGOT are annotated JAK inhibitors.

These results suggest that KG-based methods and DRIAD-SP are complementary. KG methods identify novel compound connections to established AD-related nodes in existing knowledge networks, while DRIAD-SP identifies drugs whose molecular signatures predict disease progression in patient tissues, independent of prior disease annotations. Notably, DRIAD-SP's identification of JAK inhibitors, which were largely absent from KG predictions, has motivated an ongoing clinical trial (NCT05189106).

Fourth, the authors seem to use a ridge rather than lasso regularization, and this limits mechanistic interpretability and feature selection ability.

We agree with the author. However, we previously addressed why we did not use lasso regularization in our method section (see line 1022 - 1024). Our goal was to assess whether the collective perturbation of a drug predicts disease progression, rather than to identify a sparse subset of the most predictive perturbed genes. We deliberately preselected genes based on drug perturbation profiles, and we wanted the model's performance to reflect contributions from the entire gene set. The sparsity-inducing nature of lasso regularization would exclude genes that were deliberately preselected for inclusion in the model.

Reviewer #6 (Remarks to the Author):

Thank you for your time and input! Please see our comments and edits above.

Nature Communications Response

REVIEWERS' COMMENTS

Reviewer #2 (Remarks to the Author)

All of my concerns in the previous rounds of reviews have been addressed. I believe that the data in this paper support the conclusions, and adds interesting and significant contributions to the field. Thank you to the authors for their thoughtful consideration of my critiques.

Thank you very much for your constructive input that helped us improve our manuscript. We appreciate your feedback and are pleased to know that we could address all points.

Reviewer #5 (Remarks to the Author)

I was asked to review the computational component of the paper and I find the addressing of their comments still somewhat lacking although it does not seem to be a large part of their paper.

---I asked them to write their method in its entirety, instead they left the short description in their methods section. I would appreciate an algorithm box or something of this sort.

---The method still seems ad hoc to me, maybe they can give more information on all drug perturbation data they used in DRIAD and which other drugs gave significant results. Information on specificity and precision vs recall should be given.

---Since their paper depends on experimental results more than computational results, maybe there is an orthogonal way of deriving baricitinib and ruxolitinib for testing.

I appreciate the authors compared to knowledge graph methods.

Thank you very much for your input that helped us improve our manuscript. We have provided all the details in the methods section to reproduce the work. All other drug perturbation data was provided in the first DRIAD paper (Rodriguez, et al, Nature Communications, 2021). In both this paper and the Rodriguez, et al, Science Translational Medicine, 2021 paper, the orthogonal bench science data is presented to support the hypothesis of baricitinib and ruxolitinib for therapeutic testing.

Reviewer #6 (Remarks to the Author)

Thank you for participating in this review.